# DNR-Pruning: Sparsity-Aware Pruning via Dying Neuron Reactivation in Convolutional Neural Networks

**Boyuan Wang**                                                      *b.wang16@lancaster.ac.uk*
*LIRA Centre*
*University of Lancaster*

**Richard Jiang**                                                      *r.jiang2@lancaster.ac.uk*
*LIRA Center*
*University of Lancaster*

**Reviewed on OpenReview:** *https://openreview.net/forum?id=ymUjGCNPYa*

## Abstract

In this paper, we challenge the conventional view of dead neurons—neurons that cease to activate—during deep neural network training. Traditionally regarded as problematic due to their association with optimization challenges and reduced model adaptability over training epochs, dead neurons are often seen as a hindrance. However, we present a novel perspective, demonstrating that they can be effectively leveraged to enhance network sparsity. Specifically, we propose DNR-Pruning, dying neuron reactivation based sparsity-aware pruning approach for convolutional neural networks (CNNs) that exploits the behavior of individual neurons during training. Through a systematic exploration of hyperparameter configurations, we show that dying neurons can be harnessed to improve pruning algorithms. Our method dynamically monitors the occurrence of dying neurons, enabling adaptive sparsification throughout CNN training. Extensive experiments on diverse datasets demonstrate that DNR-Pruning outperforms existing sparsity-aware pruning techniques while achieving competitive results compared to state-of-the-art methods. These findings suggest that dying neurons can serve as an efficient mechanism for network compression and resource optimization in CNNs, opening new avenues for more efficient and high-performance deep learning models.

## 1 Introduction

The exponential rise in AI power consumption Crawford (2024), driven by increasing data volumes and model sizes, calls for efficient solutions to meet net-zero goals. Neural network pruning addresses this challenge by reducing model size, memory usage, and computational costs, enabling faster inference and lower energy consumption Shinde (2024); Fang (2024). It supports deployment on resource-constrained devices, enhances generalization, and optimizes hardware efficiency. Pruning is key to sustainable AI development, advancing scalable, environmentally responsible solutions.

Recent advancements have extensively explored pruning methods to enhance the efficiency of deep neural networks across various domains. While pruning has been widely applied to CNNs LeCun (2015); Zhao (2024); Wu (2024), recent research has extended its scope to other architectures, such as Transformers Pope (2023); Fang (2022); Sun (2024) and spiking neural networks (SNNs) Shi (2024); Deng (2021); Chen (2022b). Notably, pruning methodologies differ across these architectures due to their distinct computational paradigms—Transformers emphasize attention mechanisms, whereas SNNs prioritize biologically inspired event-driven processing, fundamentally diverging from artificial neural networks (ANNs). In this paper, we focus on CNNs, as they provide a well-established benchmark with a broader range of competitive pruning strategies.

Pruning has emerged as a fundamental technique for reducing the computational cost of deep neural networks while maintaining competitive performance. Existing CNN pruning methods can be broadly categorized into sparsity-aware and non-sparsity-aware approaches. Sparsity-aware methods explicitly analyze the impact of different sparsity levels on test accuracy, facilitating comparisons across varying degrees of parameter reduction. Notable works in this domain include entropy-guided pruning for depth reduction Liao (2023), early-stage pruning based on the lottery ticket hypothesis Rachwan (2022), and iterative ranking of sensitivity statistics Verdenius (2020). Additionally, studies on data-agnostic pruning at initialization Pham (2023) and large-scale pruning as a discrete optimization problem Chen (2022a) underscore the role of sparsity in optimizing network performance. In contrast, non-sparsity-aware methods focus on structural pruning without explicitly considering the accuracy trade-offs across different sparsity levels. Representative works include entropy-based pruning for transfer learning under extreme resource constraints Spadaro (2023), graph-based structural pruning frameworks Fang (2023), dynamic structure pruning Park (2023), and structural alignment via partial regularization Gao (2023). While these approaches enhance model efficiency, they often overlook the role of neuron inactivity in performance degradation. In contrast, our method reexamines dying neurons by dynamically integrating entropy and Kullback-Leibler (KL) divergence as measures of dying neuron reactivation, thereby evaluating dying neurons for sparsity-aware pruning during the training of CNNs. This enables real-time evaluation and pruning of neurons, optimizing model performance while simultaneously reducing redundancy, rather than relying solely on structural optimization or post-training pruning.

Dead neurons are often viewed as detrimental to neural networks, leading to suboptimal performance Maas (2013); Douglas & Yu (2018) or reduced adaptability Abbas (2023); Lyle (2023), especially in dynamic environments. To mitigate this, alternative activation functions that prevent saturation, such as Leaky ReLU Maas (2013), Parametric ReLU Tao & Thakur (2024), ELU Kim (2020), Swish Ramachandran (2017), and GELU Peltekis (2024), have been proposed to implement various CNNs. In this work, we broaden the concept of "dead neurons"—which has predominantly been associated with activation function issues—to a more general notion of "dying neurons" that reflects challenges present throughout the neural network. Specifically, unlike a dead neuron that is permanently inactive, a dying neuron is not fully incapacitated and may potentially reactivate in subsequent training epochs. Moreover, we introduce the concept of Dying Neuron Reactivation (DNR) as a novel aspect of this paper, emphasizing its potential role in mitigating performance degradation and enhancing network adaptability. We examine this phenomenon through the lenses of network sparsity and pruning, thereby laying the groundwork for novel approaches to developing more efficient and compact neural architectures.

We introduce **Dying Neuron Reactivation Based Sparsity-Aware Pruning (DNR-Pruning)**, a novel approach that exploits neuron inactivity to optimize network sparsity. By adjusting some training hyperparameters—such as learning rate, batch size, and regularization—we analyze their impact on neuron death. DNR-Pruning dynamically removes dying neurons during optimization, improving both efficiency and convergence. Extensive experiments on diverse datasets demonstrate that our method outperforms over 10 state-of-the-art methods. Notably, pruning at various sparsity levels enhances accuracy while maintaining competitive compression trade-offs. As deep learning models scale, DNR-Pruning provides an energy-efficient solution aligned with sustainable AI development.

## 2 Related Work

### 2.1 CNN Pruning

Pruning techniques have been extensively studied to improve CNN efficiency by reducing parameter count and computational cost. Beyond the conventional structured and unstructured taxonomy, existing methods can also be categorized into sparsity-aware pruning—which explicitly accounts for the impact of varying sparsity levels on model accuracy—and non-sparsity-aware pruning, which performs structural compression without directly considering sparsity-dependent performance variations. We emphasize that this orthogonal categorization offers a complementary perspective for understanding pruning strategies.

Sparsity-aware pruning methods analyze model performance across different sparsity levels, providing insights into the trade-offs between compression and accuracy. Liao et al. introduce the Entropy Guided Pruning (EGP)

algorithm, which prioritizes the pruning of connections in layers exhibiting low entropy Liao (2023). Rachwan et al. propose an early-stage pruning strategy inspired by the lottery ticket hypothesis, demonstrating that certain subnetworks can be identified and pruned efficiently before full training Rachwan (2022). Verdenius et al. introduce a sensitivity-based ranking approach, iteratively selecting parameters for pruning based on their impact on network performance Verdenius (2020). Pham et al. explore data-agnostic pruning at initialization, emphasizing the importance of designing effective sparse masks without reliance on training data Pham (2023). Chen et al. frame pruning as a discrete optimization problem, proposing large-scale network pruning techniques that efficiently balance sparsity and accuracy Chen (2022a). These methods collectively demonstrate that sparsity-aware strategies enable a more performance-preserving reduction of model complexity.

Non-sparsity-aware pruning methods focus on structural optimization without explicitly evaluating accuracy variations across different sparsity levels. Spadaro et al. propose the Efficient Neural Transfer with Reduced Overhead via Pruning (ENTROPI) algorithm, an entropy-based pruning method designed for transfer learning under stringent memory and computation constraints Spadaro (2023). Fang et al. propose DepGraph, a graph-based approach for structural pruning that enables flexible network compression across various architectures Fang (2023). Park et al. develop a dynamic structure pruning method that adaptively removes redundant structures based on training dynamics Park (2023). Gao et al. employ a structural alignment technique with partial regularization to optimize network sparsity while maintaining generalization performance Gao (2023).

Despite the progress in both sparsity-aware and non-sparsity-aware pruning methods, the issue of dying neurons—neurons that become inactive during training and contribute little to network expressivity—remains underexplored. Traditional pruning methods primarily focus on weight sparsity while neglecting the role of neuron inactivity in performance degradation. This motivates our work, where we introduce a sparsity-aware pruning framework that systematically identifies and removes dying neurons at different sparsity levels. By integrating neuron activity into the pruning process, our method mitigates the negative effects of dying neurons while optimizing network efficiency.

## 2.2 Dead Neuron

The ReLU activation function, though beneficial in avoiding gradient vanishing and gradient saturation during backpropagation, suffers from the "dead neuron" problem, wherein neurons can become permanently inactive if the learning rate is too high, causing weight updates that make neurons unresponsive to subsequent activations Evci (2020); Lasby (2024); Lee (2024b); Whitaker & Whitley (2023); Jiang (2022). This occurs when $w' = w - \eta \Delta w$ results in a negative $w'$ due to a large learning rate $\eta$, leading to an output of zero and zero gradient for ReLU, thus halting further updates. Dead neurons are a persistent issue, especially when network initialization is suboptimal Sokar (2023); Abbas (2023). Proposed solutions include modifying activations using adaptive mechanisms, such as ConvReLU or Leaky ReLU Gao (2020); Lu (2020), which introduce mechanisms to address the problem in the early layers or dynamically adjust activation thresholds, thereby maintaining the network's ability to learn effectively. Additionally, Liu et al. introduced the "Activate-while-Pruning (AP)" method, which selectively activates neurons during the pruning process to reduce the dynamic dead neuron rate, thereby enhancing the performance of pruned networks Liu (2023).

In traditional literature, the term "dead neuron" has primarily described issues localized to activation functions—specifically, neurons that become inactive due to negative inputs or inappropriate thresholding, resulting in zero gradients and irreversible inactivity. In contrast, our work introduces the concept of the "dying neuron" as a broader, system-level phenomenon that permeates the entire network. Unlike dead neurons, dying neurons exhibit transient inactivity; they are not irreversibly inoperative and may reactivate in subsequent training epochs. This nuance is essential for a more accurate diagnosis of network inefficiencies and the development of targeted remediation strategies. Furthermore, we propose a novel framework termed Dying Neuron Reactivation (DNR). DNR is designed to identify neurons that, while currently dormant, retain the potential to resume activity and contribute to the network's performance. By leveraging DNR, we can dynamically monitor dying neurons, thereby preserving valuable network capacity and informing more effective pruning strategies. This approach not only enhances the robustness of the network but also opens new avenues for optimizing training dynamics and overall performance. By re-conceptualizing neuron

inactivity from a static "dead" state to a dynamic "dying" process—with the possibility of reactivation through DNR—we provide a more comprehensive framework for understanding and addressing neural degradation.

## 3 Method

In this section, we explore the phenomenon of accumulating dying neurons during CNN training. We provide theoretical insights into how training heuristics and hyperparameters influence neuron death and propose DNR-Pruning, a novel dying neuron reactivation based sparsity-aware pruning method. DNR-Pruning jointly considers both entropy reduction and KL divergence increase to assess neuron redundancy. This dual criterion ensures that neurons selected for removal are not just low-variance but also statistically uninformative relative to the data distribution, effectively identifying genuinely inactive neurons. By monitoring DNR in neuron outputs, DNR-Pruning identifies underperforming neurons for pruning. This approach enhances model efficiency without the need for pre-training pruning steps, streamlining the training process while maintaining performance.

### 3.1 Dying Neuron Analysis

In a deep neural network with $N$ training instances, the response profile of the $k$-th neuron in layer $m$ is defined as $r_{mk} \in \mathbb{R}^N$. We refer to "dying neurons" as those exhibiting diminished or stagnant activations in this context.

**Definition 3.1** (**Dying Neuron**). A neuron in the $m$-th layer is regarded as inactive if its response $r_{mk} = 0$ holds consistently across the entire training dataset. Neurons that enter this inactive state and remain so throughout training are identified as dying.

Many contemporary network architectures make use of activation functions that include saturation zones with zero values. In these scenarios, once a neuron enters an inactive state during training, its incoming weights often experience gradients that are zero or very close to zero, making it challenging for the neuron to re-engage. In this work, we primarily utilize the ReLU activation function, defined as $f(x) = \max(0, x)$, where neuron activation is determined by the sign of the pre-activation input.

The network parameters are represented by $\theta$, optimized to reduce the overall training error $\mathcal{L}(\theta) = \frac{1}{N} \sum_{j=1}^{N} \ell_j(\theta)$, where $\ell_j(\theta)$ represents the individual loss for the $j$-th sample. This process employs techniques derived from SGD. At each step, the loss gradient $\nabla \mathcal{L}(\theta)$ is approximated using a randomly selected mini-batch $B \subset \{1, \ldots, N\}$. For SGD with learning rate $\alpha$, the update step is given by:

$$\theta_{t+1} = \theta_t - \alpha \widehat{\nabla \mathcal{L}}(\theta_t, B_t), \quad \widehat{\nabla \mathcal{L}}(\theta, B) := \frac{1}{|B|} \sum_{j \in B} \nabla \ell_j(\theta) \tag{1}$$

We begin with some empirical observations. Utilizing the aforementioned definition, we monitored the dying neurons when training a ResNet-18 on CIFAR-10 using the SGD optimizer with different learning rates and activation functions. We employed a negative slope of $\alpha = 0.05$ for Leaky ReLU, and $\beta = 1$ for Swish. The result is explained in the appendix.

**Influence of Noise**. Although not all dying neurons are caused by noise, we hypothesize that noise during training contributes significantly to neuron inactivity Vivien (2022). To test this, we trained a ResNet-18 model on CIFAR-10, isolating noise from single-sample gradient estimates $\widehat{\nabla \mathcal{L}}(\theta_{mj})$ and reconstructing the full gradient $\nabla \mathcal{L}(\theta_{mj})$ for analysis. Figure 1 shows that noise leads to the accumulation of dying neurons, highlighting that not all dying neurons are due to gradients pointing to dying zones. We also compare different noise regimes to emphasize the impact of SGD's noise structure on neuron death.

**Learning rate and batch size**. Our model reveals a relationship between learning rate, batch size, and neuron death: by affecting noise variance in optimizer updates, both hyperparameters influence the proportion of dying neurons Keskar (2017); Smith (2018; 2017). This prediction is supported by Figure 2. A detailed exploration of these hyperparameter effects on neuron death is provided in the appendix.

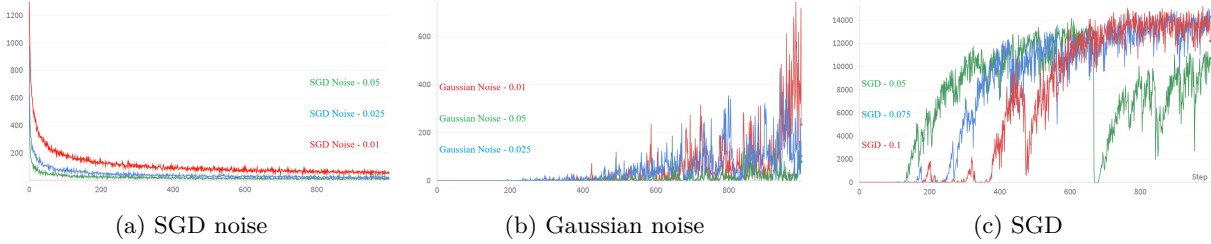

Figure 1: Training a ResNet-18 on CIFAR-10: (a) SGD Noise. Only the noisy component of the mini-batch gradient is applied, showing that noisy updates alone can cause neurons to become inactive post-initialization. This selective effect suggests noise impacts only active neurons, with dying neurons unaffected by subsequent updates. (b) Gaussian Noise. Unlike SGD noise, Gaussian noise leads to a steady rise in dying neurons, particularly in later training stages. Gaussian noise lacks a mechanism to revive neurons, creating a cumulative effect on neuron inactivity. (c) Standard SGD. During standard SGD training, there is an initial rapid increase in dying neurons under noise. As training proceeds and convergence nears, the rate of neuron death stabilizes, suggesting the model reaches a state where noise no longer significantly impacts neuron activity.

**Regularization**. Regularization limits the solution space for machine learning optimizers by encouraging smaller parameter magnitudes, effectively pulling solutions closer to the origin Pesme (2021). In ReLU networks, neurons become inactive when $\theta_{mj} = 0$ or when all weights are negative, leading to $\text{ReLU}(\theta_j \cdot x_l^T) = 0$ (with $x_{il} \geq 0$). While the exact inactivity boundary is unclear, proximity to the origin increases the likelihood of neuron inactivity.

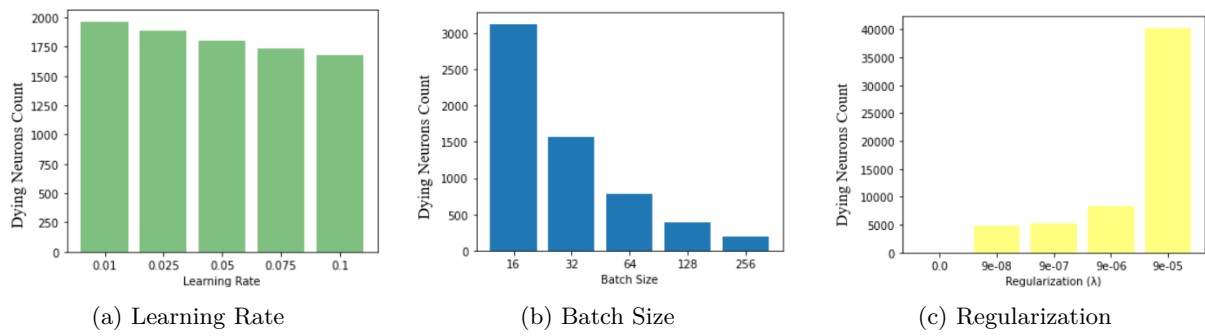

Figure 2: For ResNet-18 on CIFAR-10, adjusting hyperparameters affects the number of dying neurons. We vary around the combination of learning rate 0.05, batch size 128, and $\lambda = 0$. For regularization, we start from the combination of learning rate 0.005, batch size 128, and $\lambda = 9 \times 10^{-8}$. For batch size variations, we maintain the same number of training steps for fair comparison.

## 3.2 Dying Neuron Reactivation Based Pruning Method

**Disclaimer.** This section combines both rigorous and heuristic elements. In particular, the formal *definitions* and *lemmas* are stated with mathematical precision and should be regarded as rigorous results. In contrast, the entropy and KL-based derivations are intended as heuristic interpretations that provide intuition and motivation for the proposed pruning framework, rather than strict theoretical proofs. We highlight this distinction so that readers are aware before delving into the subsequent theoretical results.

In deep neural networks, each neuron's activity is characterized not only by its response to input data but also by the evolution of its weight distribution during training. Assuming that the input to a neuron follows a probability distribution $P(x)$ (with $x = [x_1, x_2, \dots, x_n]$ representing input features), and that the neuron's

activation is given by

$$y = f\left(\sum_{j=1}^{n} w_{i,c,j}^{(t)} x_j + b_{i,c}^{(t)}\right), \tag{2}$$

this transformation induces an output distribution $Q(y)$, shaped by both the distribution of pre-activations and the nonlinearity $f(\cdot)$. The nonlinear function can amplify or suppress variations in the input, making the output entropy a potentially informative proxy of neuron activity. High output entropy suggests dynamic and flexible responses, whereas low entropy implies saturated or inactive behavior, often described as "dying" neurons Hazarika (2018); Ghiasi-Shirazi (2019).

To better understand the interplay between input variability and weight magnitude, we define the contribution of each input dimension $x_j$ to the activation of neuron $c$ in layer $i$ at epoch $t$ as:

$$c_j = \left|w_{i,c,j}^{(t)}\right| \cdot \sigma(x_j), \tag{3}$$

where $\sigma(x_j)$ denotes the standard deviation of input feature $x_j$. This formulation emphasizes dimensions with both high input variability and strong weight coupling. A normalized distribution over input contributions is then given by:

$$p(x_j) = \frac{c_j}{\sum_{k=1}^{n} c_k} = \frac{\left|w_{i,c,j}^{(t)}\right| \cdot \sigma(x_j)}{\sum_{k=1}^{n} \left|w_{i,c,k}^{(t)}\right| \cdot \sigma(x_k)}. \tag{4}$$

Although this contribution distribution simplifies the complex relationship between inputs and outputs, it provides a heuristic means to examine the diversity of neuron responses. Based on this, we define the entropy of the neuron's output contribution as:

$$H_{\text{out}}^{(t)}(y) = -\sum_{j=1}^{n} p(x_j) \log_2\left(p(x_j)\right). \tag{5}$$

Importantly, this entropy does not directly serve as a pruning score. Instead, it offers conceptual motivation: neurons with low output entropy likely contribute little due to weak input coupling or activation saturation, prompting us to investigate their internal parameter dynamics.

To that end, we track the evolution of each neuron's weight distribution over time. Let $W_{i,c}^{(t)} = \{w_{i,c,1}^{(t)}, w_{i,c,2}^{(t)}, \ldots, w_{i,c,n}^{(t)}\}$ denote the set of incoming weights to neuron $c$ in layer $i$ at epoch $t$. We define the weight entropy as:

$$H(W_{i,c}^{(t)}) = -\sum_{j=1}^{n} p(w_{i,c,j}^{(t)}) \log_2\left(p(w_{i,c,j}^{(t)}) + \epsilon\right), \tag{6}$$

where $p(w_{i,c,j}^{(t)})$ is a normalized distribution over weight magnitudes and $\epsilon$ is a small constant added for numerical stability. The change in weight entropy across epochs is defined as:

$$\Delta H_{i,c}^{(t)} = H(W_{i,c}^{(t-1)}) - H(W_{i,c}^{(t)}), \tag{7}$$

which measures how concentrated the weight distribution becomes. Additionally, we evaluate the Kullback-Leibler (KL) divergence between weight distributions at successive epochs:

$$D_{\text{KL}}\left(W_{i,c}^{(t)} \parallel W_{i,c}^{(t-1)}\right) = \sum_{j=1}^{n} p(w_{i,c,j}^{(t)}) \log\left(\frac{p(w_{i,c,j}^{(t)})}{p(w_{i,c,j}^{(t-1)}) + \epsilon}\right), \tag{8}$$

which captures abrupt shifts in the neuron's weight landscape. A large entropy drop ($\Delta H_{i,c}^{(t)} > 0$) combined with a rise in KL divergence signals a reduction in weight diversity—an indicator of potential neuron inactivity or saturation.

In summary, while the output entropy $H_{\text{out}}^{(t)}(y)$ highlights the functional stagnation of a neuron due to activation saturation or ineffective connectivity, our pruning strategy is grounded in the temporal dynamics of the weight distribution. This approach allows us to distinguish between structurally stagnant neurons and those with evolving, but currently underutilized, roles.

### 3.2.1 Dying Neuron Reactivation

While our approach primarily focuses on pruning, the term Dying Neuron Reactivation (DNR) is intended to highlight a key conceptual distinction from conventional pruning techniques. Specifically, DNR does not rely on a static criterion to permanently remove neurons. Instead, it adopts a dynamic perspective—monitoring neuron behavior over time using entropy and KL divergence to identify neurons that appear inactive (or "dying") at certain stages of training but may later regain importance as the learning dynamics evolve. In this sense, while we do not explicitly "reactivate" neurons in the traditional sense, our method implicitly allows for reactivation by avoiding premature or irreversible pruning decisions. This temporal flexibility is crucial for maintaining the adaptability of the network throughout training. Rather than relying solely on entropy or KL divergence, we propose the DNR metric as a unified measure to assess the reactivation potential of dying neurons. The DNR for neuron $c$ in layer $i$ at epoch $t$ is defined as follows:

**Definition 3.2** (Dying Neuron Reactivation). For neuron $c$ in layer $i$ at epoch $t$, the Dying Neuron Reactivation (DNR) is defined by:

$$\text{DNR}_{i,c}^{(t)} = \alpha \cdot \frac{\Delta H_{i,c}^{(t)}}{H(W_{i,c}^{(t-1)}) + \epsilon} + \beta \cdot \frac{D_{KL}(W_{i,c}^{(t)} \parallel W_{i,c}^{(t-1)})}{D_{\max} + \epsilon},$$

where:

- $\Delta H_{i,c}^{(t)} = H(W_{i,c}^{(t-1)}) - H(W_{i,c}^{(t)})$ represents the reduction in weight entropy;

- $\alpha$ and $\beta$ are hyperparameters that balance the contributions from entropy reduction and KL divergence, respectively;

- $D_{\max}$ is the maximum KL divergence observed during the current training phase (or an alternative normalization factor);

- $\epsilon$ is a small constant to avoid division by zero.

A higher value of $\text{DNR}_{i,c}^{(t)}$ indicates that neuron $c$ is undergoing significant entropy reduction along with increased KL divergence, suggesting that the neuron is progressively becoming redundant.

**Definition 3.3** (Dying Neuron Reactivation Threshold). To differentiate between essential neurons and those that can be pruned, we introduce the Dying Neuron Reactivation Threshold $\tau$. If a neuron $c$ in layer $i$ at epoch $t$ satisfies

$$\text{DNR}_{i,c}^{(t)} > \tau,$$

it is considered to be in a low-activity or "reactivating" state and is marked as a candidate for pruning. The threshold $\tau$ can be determined empirically or tuned via cross-validation.

**Definition 3.4** (Dying Neuron Reactivation Potential). The potential for a neuron to recover from a low-activity state is quantified by the Dying Neuron Reactivation Potential $R_{i,c}$, defined as:

$$R_{i,c} = 1 - \frac{\text{DNR}_{i,c}^{(t)}}{\max_t \{\text{DNR}_{i,c}^{(t)}\} + \epsilon},$$

with $R_{i,c} \in [0, 1]$. A value of $R_{i,c}$ close to 0 indicates persistent high DNR (i.e., high redundancy), whereas larger values imply that the neuron retains a significant potential to recover.

**Lemma 3.5** (High DNR Indicates Redundant Information Encoding)**.** *If, over successive epochs, a neuron c consistently exhibits a substantial reduction in weight entropy ($\Delta H_{i,c}^{(t)} > 0$) alongside increased KL divergence—resulting in a high $DNR_{i,c}^{(t)}$—then the neuron's weight distribution becomes increasingly concentrated. This shift implies reduced variability in its representational capacity, suggesting that the neuron's contribution becomes redundant with respect to the learned representation, and can therefore be pruned without loss of expressivity.*

**Lemma 3.6** (DNR Stability is Independent of Initial Weight Distribution)**.** *Under uniform normalization conditions, the computed $DNR_{i,c}^{(t)}$ for any neuron c in layer i remains invariant with respect to the initial weight distribution. This invariance is due to the normalization terms $\frac{\Delta H_{i,c}^{(t)}}{H(W_{i,c}^{(t-1)})+\epsilon}$ and $\frac{D_{KL}(W_{i,c}^{(t)}\|W_{i,c}^{(t-1)})}{D_{\max}+\epsilon}$, which eliminate absolute scale effects, ensuring that the DNR metric captures only the relative changes over time.*

**Definition 3.7** (DNR-Driven Sparsity Level)**.** For a given neural network layer $i$, the sparsity level $S_i$ is defined as the fraction of neurons identified for pruning:

$$S_i = \frac{\sum_c I\left(\text{DNR}_{i,c}^{(t)} > \tau\right)}{C_i},$$

where $I(\cdot)$ is the indicator function and $C_i$ is the total number of neurons in layer $i$.

**DNR-Pruning Summary.** In summary, the proposed dying neuron reactivation based pruning method leverages the DNR metric—a composite measure that encapsulates both the reduction in weight entropy and the increase in KL divergence—to dynamically prune dying neurons during training. This approach accelerates the training process by focusing on neurons that are becoming less informative, while simultaneously preserving model performance through adaptive, in-training pruning. Such dynamic pruning promotes a sparser network architecture without compromising accuracy Cohen (2021); Wang (2021); Rachwan (2022).

# 4 Experiments

Our experimental focus lies on computer vision tasks. We trained MLPNet, ResNet-18 and VGG-16 networks on MNIST, CIFAR-10 and Tiny-ImageNet. Following the training scheme outlined by Evci et al Evci (2020). For the ResNet and VGG architecture, we expanded the scope of our experiments Rachwan (2022). We used 30G bytes of memory, an NVidia V100 GPU and an Intel(R) Xeon(R) Platinum 8352Y CPU @ 2.20GHz CPU. And more experimental results are presented in the appendix.

Our approach involves a form of neural "inactivation" or "disabling," where pruning occurs during training, transitioning networks from dense to sparse. The methods we compare against operate within a similar paradigm. We compare our approach with existing methods, including Crop-it/EarlyCrop Rachwan (2022), SNAP Verdenius (2020), and a modified version of the early pruning strategy from Rachwan et al. Rachwan (2022), referred to as EarlySNAP. These methods are trained using the configurations recommended by their authors, without incorporating the regularization regime used in our method. Across all comparisons, our approach achieves comparable or superior performance.

The results across Tables 1 to 4 demonstrate that DNR-Pruning consistently outperforms other pruning methods by maintaining higher accuracy at equivalent sparsity levels. Specifically, for ResNet-18 on CIFAR-10, DNR-Pruning achieves significantly better performance under both neural and weight sparsity settings (Tables 1 and 2), preserving accuracy even at high sparsity levels. This advantage is mirrored in VGG-16 networks (Tables 3 and 4), where DNR-Pruning maintains a stable accuracy lead, particularly notable at extreme weight sparsity values. The results confirm that DNR-Pruning's approach to optimizing sparse networks more effectively retains critical model capacity compared to other methods, suggesting that it is better suited to high-sparsity scenarios without substantial loss in accuracy.

Table 1: For ResNet-18 networks on CIFAR-10, DNR-Pruning can find sparser solutions maintaining better performance than other approaches, including Cropit, EarlyCrop and EarlySNAP Rachwan (2022), SNAP Verdenius (2020). **Neural sparsity (%)**.

|  | 50 | 60 | 70 | 75 | 80 | 85 | 90 |
|---|---|---|---|---|---|---|---|
| CroPit-S | 92.15 | 91.18 | 90.98 | 90.00 | 88.90 | 88.10 | 85.10 |
| EarlyCroP-S | 92.33 | 92.23 | 91.75 | 91.35 | 90.78 | 87.85 | 84.18 |
| EarlySNAP | 92.08 | 92.33 | 92.00 | 91.25 | 90.43 | 88.60 | 83.50 |
| SNAP | 91.93 | 91.48 | 91.23 | 90.48 | 89.40 | 87.55 | 85.45 |
| **DNR-Pruning** | **93.15** | **93.15** | **92.42** | **92.02** | **91.65** | **90.56** | **88.70** |

Table 2: For ResNet-18 networks on CIFAR-10, DNR-Pruning can find sparser solutions maintaining better performance than other approaches. **Weight sparsity (%)**.

|  | 75 | 80 | 85 | 90 | 95 | 97 | 98 |
|---|---|---|---|---|---|---|---|
| CroPit-S | 92.70 | 92.26 | 91.66 | 91.06 | 90.41 | 89.22 | 88.58 |
| EarlyCroP-S | 92.55 | 92.40 | 92.31 | 92.19 | 91.31 | 90.52 | 88.57 |
| EarlySNAP | 92.40 | 92.20 | 92.13 | 92.18 | 91.24 | 90.36 | 89.01 |
| SNAP | 92.19 | 91.96 | 91.74 | 91.38 | 90.80 | 89.89 | 88.83 |
| **DNR-Pruning** | **93.73** | **93.47** | **93.23** | **92.50** | **91.21** | **89.72** | **88.11** |

Table 3: For VGG-16 networks on CIFAR-10. DNR-Pruning better maintains performance at higher sparsities than other approaches. **Neural sparsity (%)**.

|  | 50 | 60 | 70 | 75 | 80 | 85 | 90 |
|---|---|---|---|---|---|---|---|
| CroPit-S | 92.22 | 92.50 | 92.25 | 92.22 | 92.00 | 91.81 | 90.89 |
| EarlyCroP-S | 89.53 | 91.77 | 92.22 | 91.94 | 91.81 | 91.80 | 90.83 |
| EarlySNAP | 89.58 | 91.81 | 92.28 | 92.22 | 92.00 | 91.66 | 77.50 |
| SNAP | 91.39 | 92.50 | 92.08 | 92.22 | 91.94 | 91.39 | 86.53 |
| **DNR-Pruning** | **92.27** | **92.53** | **92.40** | **92.33** | **92.25** | **92.12** | **91.99** |

Table 4: For VGG-16 networks on CIFAR-10. DNR-Pruning better maintains performance at higher sparsities than other approaches. **Weight sparsity (%)**.

|  | 75 | 80 | 85 | 90 | 95 | 97 | 98 |
|---|---|---|---|---|---|---|---|
| CroPit-S | 92.70 | 92.60 | 92.20 | 91.80 | 91.50 | 91.10 | 90.50 |
| EarlyCroP-S | 92.40 | 92.20 | 92.00 | 91.80 | 91.30 | 91.00 | 90.70 |
| EarlySNAP | 92.44 | 92.40 | 92.20 | 91.80 | 91.40 | 91.60 | 71.40 |
| SNAP | 92.10 | 92.00 | 92.20 | 91.80 | 90.90 | 87.30 | 78.20 |
| **DNR-Pruning** | **93.02** | **92.84** | **93.03** | **92.73** | **92.92** | **92.89** | **92.88** |

## 4.1 Experiment on Tiny-ImageNet

We included results from Pham et al.Pham (2023) in Table 5 to better illustrate the comparative performance of the state-of-the-art pruning methods against our proposed DNR-Pruning method under varying levels of weight sparsity. The results demonstrate that our DNR-Pruning approach achieves higher accuracy across varying levels of weight sparsity compared to other pruning methods.

Besides weight sparsity, we find that the FLOPs reduction of subnetwork after pruning is more important in the context of pruning before training. We have measured FLOPs of subnetworks produced by different methods in Table 5. The result indicates that our DNR-Pruning can produce subnetworks with lower FLOPs than other methods.

Table 5: For ResNet-18 networks on Tiny-ImageNet, DNR-Pruning achieves both high accuracy and low FLOPs in comparison to the state-of-the-art methods, including SNIP Lee (2019), Iterative SNIP De Jorge (2021), SynFlow Tanaka (2020), and PHEW Patil (2021) and NBP Pham (2023).

| | | Sparsity (%) | | | |
|---|---|---|---|---|---|
| | | 68.38 | 90 | 96.84 | 99 |
| Accuracy | SNIP | 56.99 | 53.43 | 48.77 | 36.02 |
| | Iter-SNIP | 56.73 | 53.60 | 48.55 | 36.42 |
| | SynFlow | 56.71 | 54.68 | 49.03 | 39.79 |
| | PHEW | 58.09 | 55.93 | 50.81 | 40.54 |
| | NPB | 58.39 | 56.82 | 51.37 | 41.05 |
| | **DNR-Pruning** | **66.96** | **61.82** | **53.70** | **41.93** |
| FLOPs ($10^8$) | SNIP | 11.35 | 5.77 | 3.04 | 1.55 |
| | Iter-SNIP | 10.73 | 7.05 | 3.98 | 1.97 |
| | SynFlow | 14.71 | 8.91 | 4.24 | 1.50 |
| | PHEW | 14.29 | 8.35 | 3.92 | 1.50 |
| | NPB | 14.37 | 5.21 | 1.74 | 0.59 |
| | **DNR-Pruning** | **5.73** | **1.81** | **0.57** | **0.18** |

Table 6: For MLPNet networks on MNIST, DNR-Pruning again outperforms the other pruning approachs, including MP, WF and CBS Yu (2022), SSP and MSP Chen (2022a). **Weight sparsity (%)**.

| | 50 | 60 | 70 | 80 | 90 | 95 | 98 |
|---|---|---|---|---|---|---|---|
| MP | 93.93 | 93.78 | 93.62 | 92.89 | 90.30 | 83.64 | 32.25 |
| WF | 94.02 | 93.82 | 93.77 | 93.57 | 91.69 | 85.54 | 38.26 |
| CBS | 93.96 | 93.96 | 93.98 | 93.90 | 93.14 | 88.92 | 55.45 |
| SSP | 93.97 | 93.94 | 93.80 | 93.59 | 92.46 | 88.09 | 46.25 |
| MSP | 95.97 | 95.93 | 95.89 | 95.80 | 95.55 | 94.70 | 90.73 |
| **DNR-Pruning** | **97.12** | **97.11** | **97.06** | **96.99** | **96.94** | **96.74** | **96.72** |

## 4.2 Experiment on MNIST

The result presented in Tables 6 demonstrates the consistent superiority of DNR-Pruning over other pruning methods across varying levels of weight sparsity for MLPNet on MNIST. For MLPNet, DNR-Pruning achieves the highest accuracy at all sparsity levels, maintaining robust performance even under extreme sparsity, significantly outperforming other methods Chen (2022a). The result underscores the efficacy of DNR-Pruning in preserving accuracy under constrained model sizes, highlighting its potential for real-world applications where memory and computation are limited.

## 4.3 Experiment on ImageNet

### 4.3.1 ResNet-56 on ImageNet

Based on the results in Table 7, our proposed method, DNR-Pruning, achieves the highest Top-1 accuracy among all compared pruning approaches on ResNet-56 with ImageNet, including a recent SOTA method published at WACV 2024. Notably, it reaches this performance within only 90 training epochs, significantly reducing the computational cost compared to methods like Hrank and AdaptDCP. While some methods such as SCP and NuSPM also achieve competitive accuracy, they either lack pre-training or demand longer training schedules. In contrast, DNR-Pruning leverages a dynamic, DNR-based criterion to prune the network effectively at initialization, maintaining high accuracy with minimal training overhead. These results demonstrate the efficiency and robustness of DNR-Pruning in large-scale image classification tasks, highlighting its advantage in both performance and training efficiency.

Table 7: Comparison of pruning methods for ResNet-56 on ImageNet. DNR-Pruning, with a sparsity of 25%, achieves the best Top-1 accuracy with fewer training epochs in comparison to the SOTA methods, including GAL Lin (2019), Hrank Lin (2020), SSS Huang & Wang (2018), Taylor Molchanov (2019), C-SGD Ding (2019), DSA Ning (2020), Hinge Li (2020), AdaptDCP Zhuang (2018), SCP Kang & Han (2020), PaS Li (2022), PaT Wang (2020), and NuSPM Lee (2024a). "PT" represents requiring a pre-trained network, and "Y" and "N" mean yes or no, respectively.

| Method | PT | Top-1 Acc. (%) | Epochs |
|---|---|---|---|
| Baseline (Unpruned) | Y | 76.5 | 90 |
| GAL | Y | 72.0 | 150 |
| Hrank | Y | 75.0 | 570 |
| SSS | N | 71.8 | 100 |
| Taylor | Y | 74.5 | * |
| C-SGD | Y | 74.9 | * |
| DSA | N | 74.7 | 120 |
| Hinge | Y | 74.7 | * |
| AdaptDCP | Y | 75.2 | 210 |
| SCP | N | 75.3 | 200 |
| PaS | Y | 74.5 | 150 |
| PaT | N | 74.9 | 90 |
| NuSPM | N | 75.3 | 90 |
| **DNR-Pruning** | **Y** | **76.0** | **90** |

Table 8: Comparison of pruning methods for MobileNet-V2 on ImageNet. DNR-Pruning achieves the best Top-1 accuracy (%) in comparison to the SOTA methods, including POT Lazarevich (2021), RigL Evci (2020), STR Kusupati (2020), UniPTS Xie (2024).

| | 50 | 60 | 70 | 80 | 90 |
|---|---|---|---|---|---|
| POT | 69.25 | 63.39 | 47.07 | 9.13 | 0.20 |
| RigL | 66.57 | 64.75 | 60.72 | 52.19 | 30.44 |
| STR | 30.96 | 20.57 | 14.62 | 9.40 | 5.32 |
| UniPTS | 69.80 | 68.01 | 64.93 | 59.47 | 42.46 |
| **DNR-Pruning** | **70.11** | **68.23** | **66.75** | **62.77** | **46.29** |

### 4.3.2 MobileNet-V2 on ImageNet

Table 8 presents a comprehensive comparison of Top-1 accuracy across various sparsity levels for MobileNet-V2 on the ImageNet dataset, evaluating several SOTA pruning methods. Notably, our proposed method, DNR-Pruning, consistently outperforms all methods across all sparsity regimes. At moderate sparsity, DNR-Pruning achieves the highest accuracy, surpassing strong pruning methods such as UniPTS and RigL. For instance, at 70% sparsity, DNR-Pruning maintains a Top-1 accuracy of 66.75%, outperforming UniPTS and RigL . At extreme sparsity levels, where most methods suffer severe degradation , DNR-Pruning still retains significantly better performance, highlighting its superior ability to preserve essential neurons. These results demonstrate that DNR-Pruning not only achieves SOTA performance under standard pruning ratios but also exhibits strong robustness under extreme compression, validating the effectiveness of our DNR-based selection strategy in identifying neurons critical for generalization.

### 4.4 Leaky ReLU

Our pruning method is specifically designed for CNNs, where neurons can become entirely inactive, leading to the emergence of dying neurons that can be effectively pruned. While Leaky ReLU Maas (2013) mitigates complete neuron deactivation by allowing small negative activations, it still exhibits a quasi-saturated regime wherein activations remain predominantly negative and contribute minimally to the network's predictions. We hypothesize that neurons operating primarily within this low-activation regime behave analogously to

dying neurons in ReLU networks, and therefore, they can be pruned without adversely affecting performance. To validate this hypothesis, we extend our approach to ResNet-18 trained with Leaky ReLU activations. In our experiments, we identify and remove neurons that consistently produce negative responses across representative minibatches—indicative of their functional dormancy. As presented in Tables 9 and 10, DNR-Pruning consistently outperforms alternative methods across various levels of both neural and weight sparsity, demonstrating its effectiveness even in architectures where neurons do not become completely deactivated.

Table 9: For ResNet-18 networks with *Leaky ReLU* on CIFAR-10. DNR-Pruning again outperforms the other pruning methods. **Neural sparsity (%)**.

|  | 50 | 60 | 70 | 75 | 80 | 85 | 90 |
|---|---|---|---|---|---|---|---|
| EarlyCroP-S | 88.89 | 88.97 | 88.17 | 87.10 | 85.99 | 84.72 | 80.71 |
| EarlySNAP | 89.40 | 87.86 | 87.02 | 85.99 | 85.00 | 84.33 | 80.00 |
| SNAP | 88.21 | 87.62 | 86.67 | 86.07 | 85.48 | 81.75 | 78.33 |
| **DNR-Pruning** | **91.93** | **91.03** | **90.67** | **90.48** | **89.82** | **88.63** | **86.14** |

Table 10: For ResNet-18 networks with *Leaky ReLU* on CIFAR-10. DNR-Pruning again outperforms the other pruning methods. **Weight sparsity (%)**.

|  | 75 | 80 | 85 | 90 | 95 | 97 | 98 |
|---|---|---|---|---|---|---|---|
| EarlyCroP-S | 88.95 | 88.90 | 88.94 | 88.78 | 87.56 | 86.34 | 85.48 |
| EarlySNAP | 89.33 | 89.39 | 88.78 | 87.72 | 86.66 | 85.59 | 84.78 |
| SNAP | 88.34 | 88.22 | 87.85 | 87.29 | 86.27 | 85.39 | 83.32 |
| **DNR-Pruning** | **93.84** | **93.61** | **93.43** | **92.58** | **91.04** | **89.69** | **87.86** |

## 4.5 Ablation Study

To evaluate the individual and joint contributions of KL divergence ($\alpha$) and entropy ($\beta$) in guiding neuron death, we conducted an ablation study on the CIFAR-10 dataset using ResNet-18 under various sparsity levels. As shown in Table 11, pruning solely based on KL divergence or entropy leads to performance degradation as sparsity increases, with KL divergence only pruning slightly outperforming entropy only pruning across all settings. Incorporating both signals through a convex combination improves robustness. Notably, the hybrid configurations of $0.75\alpha + 0.25\beta$ and $0.25\alpha + 0.75\beta$ yield competitive but marginally different results, suggesting complementary behavior between the two criteria. However, our proposed DNR-Pruning, which adaptively balances the two signals (i.e., $\alpha = \beta = 0.5$), consistently achieves the highest test accuracy across all sparsity levels. For instance, at 95% sparsity, DNR-Pruning achieves 91.21%, outperforming KL divergence only and entropy only. These results demonstrate that an equal-weighted fusion of KL divergence and entropy provides a more robust and expressive signal for pruning, validating the effectiveness of our dynamic DNR-Pruning strategy.

Table 11: On the CIFAR-10 dataset, DNR-Pruning achieves higher performance under the same sparsity levels in ResNet-18 compared to other ablation-based methods. $\alpha$ is KL Divergence and $\beta$ is Entropy. **Weight sparsity (%)**.

| $\alpha$ | $\beta$ | 75 | 80 | 85 | 90 | 95 | 97 | 98 |
|---|---|---|---|---|---|---|---|---|
| 0 | 1 | 93.76 | 93.44 | 93.01 | 92.40 | 90.78 | 89.24 | 87.70 |
| 0.75 | 0.25 | 93.73 | 93.45 | 93.04 | 92.45 | 90.89 | 89.42 | 87.77 |
| **0.5** | **0.5** | **93.73** | **93.47** | **93.23** | **92.50** | **91.21** | **89.72** | **88.11** |
| 0.25 | 0.75 | 93.73 | 93.43 | 93.03 | 92.32 | 90.88 | 89.21 | 87.76 |
| 1 | 0 | 93.71 | 93.30 | 92.98 | 92.31 | 90.73 | 88.99 | 87.59 |

## 5 Discussion and Conclusion

In this work, we challenge the traditional view of "dead neurons" in deep neural network training, offering a novel perspective on how these dying neurons can contribute to network sparsity. We demonstrate that dying neurons, often seen as problematic, can be effectively leveraged to enhance pruning algorithms. Our method, DNR-Pruning, introduces a dynamic sparsification approach during training by regulating the occurrence of dying neurons through DNR metrics. This enables efficient pruning without significant accuracy loss and outperforms existing techniques, showing potential for integration into established pruning frameworks. Our experiments demonstrated consistent and overwhelmingly better results compared to over 20 different SOTA methods across various experimental setups (as shown in Table 1-8, Table 9 and Table 10).

We examine how hyperparameters such as learning rate, batch size, and regularization influence the emergence of dying neurons, and subsequently, the role of neuron inactivity in shaping sparsity patterns. Our investigation shows that DNR-Pruning effectively maintains high accuracy even at extreme sparsity levels when using Leaky ReLU activation in convolutional layers. While this paper focuses exclusively on CNNs, we acknowledge the potential of our DNR based sparsity approach to extend to more advanced architectures, such as Transformers, which employ attention-driven synaptic connectivity, and large generative models, which operate on high-dimensional token representations. However, a thorough investigation of these architectures falls beyond the scope of this work and is left for future research. In these architectures, attention mechanisms and activation functions such as GELU and Swish could enable smoother activation reduction and redundancy elimination. Extending our method to transformers and exploring their unique sparsity characteristics is a promising direction for future work.

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

# A  Appendix

## A.1  Theoretical Analysis

### A.1.1  Effect of Learning Rate and Batch Size on Dying Neurons

In the training of deep neural networks, hyperparameters such as **learning rate** and **batch size** are crucial in determining neuron activation dynamics. Specifically, low learning rates and small batch sizes can impact gradient updates, leading to an increased likelihood of certain neurons becoming "dying neurons," i.e., neurons that gradually stop activating and cease to contribute to the network's output under certain input conditions.

**(1) The Impact of Low Learning Rate on Dying Neurons.**  A lower learning rate is often applied to improve training stability and model convergence. However, with a lower learning rate, the amplitude of each weight update is smaller. Neurons with smaller initial weights or weaker initial gradients might fail to reach activation thresholds due to these minor updates, leading to diminished activity and, ultimately, neuron death.

**Definition A.1.** For a neuron $j$ at training iteration $t$, if its activation magnitude $a_j(t)$ consistently remains below a predefined threshold $\theta$, then it is classified as a **dying neuron**:

$$a_j(t) < \theta \quad \text{for } t \to \infty \Rightarrow \text{Neuron } j \text{ is classified as a dying neuron.}$$

Empirical observations indicate that lower learning rates result in a higher occurrence of such neurons.

**Lemma A.2** (**Relationship between Learning Rate and Dying Neurons**). *Let the gradient update magnitude be $\Delta W \propto -\eta \cdot \nabla W$, where $\eta$ is the learning rate. If the learning rate $\eta$ is below a minimum threshold $\eta_{min}$, then the probability of neuron $j$ dying increases:*

$$\eta < \eta_{min} \Rightarrow \Pr(\text{dying neuron } j) \uparrow.$$

*Here, $\eta_{min}$ is an empirical threshold below which neurons are significantly more likely to die.*

**(2) The Effect of Small Batch Size on Gradient Variance and Neuron Vitality.**  Small batch sizes are often chosen to improve generalization but also bring high variance in gradient estimates. This increased gradient variance can disrupt neuron activation stability, leading to certain neurons becoming inactive early in training.

**Definition A.3** (**Gradient Variance Factor Induced by Batch Size**). Let $B$ be the batch size, and define the gradient variance factor $\sigma_g(B)$ as:

$$\sigma_g(B) = \text{Var}\left(\nabla W | B\right),$$

where the gradient variance $\sigma_g(B)$ increases as batch size decreases, affecting neuron activation stability.

**Lemma A.4** (**Relationship between Batch Size and Neuron Vitality**). *For any neuron $j$, if batch size $B$ is small, then the probability of the neuron becoming inactive increases. Specifically, there exists a threshold $B_{min}$ such that:*

$$B < B_{min} \Rightarrow \Pr(\text{dying neuron } j) \uparrow.$$

**(3) Interaction between Learning Rate and Batch Size.**  In practice, learning rate and batch size are not entirely independent, and they exhibit complex interactions in influencing neuron activation dynamics. The gradient variance induced by small batch sizes can be further amplified by a low learning rate, intensifying the tendency for neurons, especially those with initially low activation, to die.

**Definition A.5** (**Combined Dying Factor of Learning Rate and Batch Size**). Define the combined dying factor $\lambda(\eta, B)$ as:

$$\lambda(\eta, B) = \eta \cdot \sigma_g(B),$$

where a lower $\lambda$ corresponds to an increased tendency for neuron death within the network.

**Lemma A.6** (**Influence of the Combined Dying Factor on Dying Neurons**). *When the combined dying factor $\lambda(\eta, B)$ is below an empirical threshold $\lambda_{min}$, the prevalence of dying neurons increases:*

$$\lambda(\eta, B) < \lambda_{min} \Rightarrow \Pr(dying\ neuron)\ increases.$$

The above definitions and lemmas illustrate how hyperparameter settings affect neuron activation dynamics and the likelihood of neuron death. By adjusting the learning rate and batch size, particularly by controlling gradient variance, it is possible to reduce the occurrence of dying neurons, thereby enhancing the network's expressiveness and training efficacy.

### A.1.2 Effect of SGD Noise on Dying Neurons

**Definition A.7** (**SGD Noise**). SGD noise refers to the variance introduced into the gradient updates due to minibatch sampling Bottou (2010). For a model parameter $\theta$, the update at iteration $t$ is given by:

$$\Delta\theta_t = -\eta \left( \nabla_\theta \mathcal{L}(\theta) + \xi_t \right),$$

where $\eta$ is the learning rate, $\nabla_\theta \mathcal{L}(\theta)$ is the gradient of the loss function, and $\xi_t \sim \mathcal{N}(0, \Sigma_t)$ is the noise term with covariance $\Sigma_t$.

**Definition A.8** (**Effective Gradient Contribution**). The effective gradient contribution of a neuron $z_i$ at layer $l$ is defined as:

$$G_i^{(l)} = \mathbb{E}[\|\nabla_{w_i^{(l)}} \mathcal{L}\|],$$

where $\mathbb{E}$ denotes the expectation over all minibatches. A neuron with $G_i^{(l)} \to 0$ over training iterations contributes negligibly to the optimization process.

**Lemma A.9** (**Impact of SGD Noise on Neuron Gradients**). *Let $g_i^{(l)}(t)$ be the gradient of the loss with respect to $w_i^{(l)}$ at iteration $t$, and let $\xi_t \sim \mathcal{N}(0, \Sigma_t)$ be the SGD noise. Then:*

$$Var(g_i^{(l)}(t)) = Var(\nabla_{w_i^{(l)}} \mathcal{L}) + \Sigma_t.$$

*For dying neurons, $Var(\nabla_{w_i^{(l)}} \mathcal{L}) \approx 0$, and thus the gradient variance is dominated by the SGD noise Smith & Le (2018):*

$$Var(g_i^{(l)}(t)) \approx \Sigma_t.$$

**Lemma A.10** (**SGD Noise Amplifies the Probability of Dying Neurons**). *For a neuron $z_i^{(l)}$, the probability of being active (non-zero output) depends on the pre-activation $a_i^{(l)} = w_i^{(l)} x^{(l-1)} + b_i^{(l)}$. Assume $a_i^{(l)} \sim \mathcal{N}(\mu_i, \sigma_i^2)$, where $\mu_i$ and $\sigma_i^2$ are affected by SGD noise. The probability of $z_i^{(l)} = 0$ under ReLU activation is:*

$$P(z_i^{(l)} = 0) = P(a_i^{(l)} \leq 0) = \Phi\left(-\frac{\mu_i}{\sigma_i}\right),$$

*where $\Phi$ is the standard normal cumulative distribution function.*

*Due to SGD noise, $\sigma_i^2$ increases, leading to a higher $P(z_i^{(l)} = 0)$, thereby amplifying the likelihood of dying neurons He (2015).*

**Lemma A.11** (**Gradient Magnitude Decay in Dying Neurons**). *For a dying neuron $z_i^{(l)}$, its gradient magnitude decays exponentially under the influence of SGD noise Jastrzębski (2018):*

$$\|\nabla_{w_i^{(l)}} \mathcal{L}\| \leq C \exp(-\alpha t),$$

*where $C$ is a constant dependent on initialization, $\alpha > 0$ is a decay rate determined by the noise scale $\Sigma_t$, and $t$ is the training iteration. This decay prevents the neuron from recovering during training.*

**Proposition A.12** (**Noise Threshold for Neuron Reactivation**)**.** *Let $\tau$ be the noise threshold for neuron reactivation. A neuron $z_i^{(l)}$ can reactivate only if:*

$$\Sigma_t < \tau,$$

*where $\tau$ depends on the distribution of $w_i^{(l)}$, $b_i^{(l)}$, and $x^{(l-1)}$. If $\Sigma_t \geq \tau$, the neuron remains in the dying state indefinitely.*

**Theorem A.13** (**SGD Noise-Induced Dying Neuron Rate**)**.** *The expected rate of dying neurons in a layer $l$, denoted by $R^{(l)}$, is given by:*

$$R^{(l)} = \int_{\mathcal{W}} P(z_i^{(l)} = 0 \mid \Sigma_t) p(w_i^{(l)}) \, dw_i^{(l)},$$

*where $\mathcal{W}$ is the weight space, and $p(w_i^{(l)})$ is the weight distribution. Higher SGD noise ($\Sigma_t$) increases $R^{(l)}$.*

### A.1.3 The relation between $w_i^{(t)}$ and $W_{i,c}^{(t)}$

To ensure clarity and consistency in the notation, we make the following definitions:

- Let $w_{i,c,j}^{(t)}$ denote the weight connecting the $j$-th neuron in layer $i-1$ to the $c$-th neuron in layer $i$ at epoch $t$.

- Let $W_{i,c}^{(t)} = \{w_{i,c,j}^{(t)}\}_j$ represent the full vector of incoming weights to neuron $c$ in layer $i$ at epoch $t$.

### A.1.4 Justification of the Weight Distribution Assumption

**Disclaimer.** The following justification does not constitute a rigorous probabilistic modeling of weights. Instead, it reflects an intuitive perspective from the author: normalized weight magnitudes are treated as empirical distributions to facilitate entropy-based analysis. The resulting entropy measure should therefore be regarded as a descriptive heuristic rather than a strict statistical property.

The computation of weight entropy $H(W_{i,c}^{(t)})$ in equation 6 relies on interpreting the normalized magnitudes of weights as a discrete probability distribution. Specifically, given a neuron $c$ in layer $i$ at epoch $t$, the weight vector $W_{i,c}^{(t)} = \{w_1, w_2, \ldots, w_n\}$ is transformed into a distribution by computing:

$$p(W_{i,c,j}^{(t)}) = \frac{|w_j|}{\sum_{k=1}^{n} |w_k|},$$

where $p(W_{i,c,j}^{(t)})$ can be interpreted as the relative contribution of the $j$-th input weight to the overall signal processing of the neuron. This normalization assumes that the absolute magnitudes of weights are meaningful proxies for their relative influence. While this approach does not impose a probabilistic generative model on the weights themselves, it provides a heuristic empirical distribution that facilitates comparison across epochs. The entropy computed from this normalized distribution serves not as a statistical estimation of an underlying latent variable, but rather as a measure of concentration or spread in the weight values.

### A.1.5 Proof of Lemma 3.5

**Disclaimer.** The proof presented here should be understood as a heuristic argument rather than a mathematically rigorous derivation. Its purpose is to provide intuition for why high DNR values correlate with redundancy in representational capacity. The reasoning relies on empirical observations and information-theoretic interpretations, which motivate—but do not formally establish—the pruning criterion.

*Proof.* Let $W_{i,c}^{(t)} = \{W_{i,c,j}^{(t)}\}_{j=1}^{d}$ denote the weights of neuron $c$ in layer $i$ at epoch $t$, and let $p(W_{i,c,j}^{(t)})$ denote an empirical estimate of their distribution. The DNR score is defined as

$$\text{DNR}_{i,c}^{(t)} = \alpha \cdot \frac{\Delta H_{i,c}^{(t)}}{H(W_{i,c}^{(t-1)}) + \epsilon} + \beta \cdot \frac{D_{KL}(W_{i,c}^{(t)} \parallel W_{i,c}^{(t-1)})}{D_{\max} + \epsilon},$$

where $\Delta H_{i,c}^{(t)} = H(W_{i,c}^{(t-1)}) - H(W_{i,c}^{(t)}) > 0$ denotes a drop in weight entropy and $D_{KL}(W_{i,c}^{(t)} \parallel W_{i,c}^{(t-1)}) > 0$ measures the divergence from the prior distribution. A consistently high DNR score implies that the neuron's weight distribution is becoming increasingly concentrated (i.e., lower entropy), while also undergoing a nontrivial distributional shift (i.e., higher KL divergence), indicating convergent specialization. This leads to reduced variability in the neuron's input-response profile and thus a lower capacity for encoding diverse or novel features. As the neuron's output becomes increasingly predictable and potentially correlated with other similarly specialized units, its representational contribution becomes redundant. Therefore, such neurons—though not necessarily silent—are statistically compressible without performance degradation, and their removal is theoretically justified. The empirical results in our study confirm that pruning high-DNR neurons preserves or even improves accuracy, substantiating DNR as a valid redundancy criterion. □

This assumption is motivated by practical observations in deep learning, where the diversity of incoming weight magnitudes often correlates with functional flexibility. A low entropy suggests domination by a few strong weights or uniform decay (saturation), both indicative of reduced information processing diversity. Thus, the entropy serves as a descriptive tool to identify neurons with concentrated, stagnant, or collapsing connectivity patterns, making it suitable for identifying candidates for pruning under our reactivation-aware framework.

### A.1.6 Proof of Lemma 3.6

**Disclaimer.** The following proof is not fully rigorous in a formal mathematical sense. It provides an intuitive explanation for why normalization removes dependence on the initial weight distribution. The argument highlights invariance properties of entropy and KL divergence but should be viewed as an interpretive justification rather than a strict theorem-proof demonstration.

*Proof.* Let $W_{i,c}^{(0)}$ denote the initial weight vector of neuron $c$ in layer $i$, sampled from an arbitrary distribution. The DNR score at epoch $t$ is defined as

$$\text{DNR}_{i,c}^{(t)} = \alpha \cdot \frac{\Delta H_{i,c}^{(t)}}{H(W_{i,c}^{(t-1)}) + \epsilon} + \beta \cdot \frac{D_{KL}(W_{i,c}^{(t)} \parallel W_{i,c}^{(t-1)})}{D_{\max} + \epsilon},$$

where $\Delta H_{i,c}^{(t)} = H(W_{i,c}^{(t-1)}) - H(W_{i,c}^{(t)})$ and $D_{\max}$ is the maximum KL divergence across all neurons and training steps. Crucially, both terms are expressed in normalized form: the entropy reduction is divided by the entropy of the previous state, and the KL divergence is scaled by a global maximum. These normalization factors render the metric invariant to the absolute magnitude or scale of the initial weight distribution. More formally, suppose $W_{i,c}^{(0)}$ is scaled or shifted within a family of distributions; since both entropy and KL divergence are information-theoretic quantities that are invariant under affine-preserving normalizations (up to additive constants canceled by the differencing or ratio), the DNR score remains determined solely by the relative change from $t-1$ to $t$. Hence, under consistent normalization, $\text{DNR}_{i,c}^{(t)}$ is unaffected by the specific choice of $W_{i,c}^{(0)}$, capturing only the dynamics of training-induced structural change. □

### A.1.7 Core Algorithm Analysis

The DNR-Pruning algorithm 1 integrates training and pruning in a unified framework to iteratively optimize both model performance and efficiency. It begins with the initialization of model parameters and a pruning mask that dictates the inclusion or exclusion of neurons during training. Over a specified number of epochs, the model is trained on the dataset using standard gradient descent updates. At predetermined pruning milestones, the algorithm dynamically adjusts the pruning mask based on DNR metrics. By alternating between training and pruning phases, the model continuously redistributes representational capacity, preserving accuracy while progressively reducing redundancy. This design ensures compatibility with resource-constrained deployment scenarios, where computational and memory efficiency are paramount.

Our pruning method integrates the reduction in neuron activation entropy and the increase in weight distribution KL divergence into a unified DNR metric, where a high DNR value identifies a neuron as

redundant and thus a candidate for pruning. During the pruning phase, the algorithm captures activation patterns for each neuron across the training dataset, constructing a probability distribution over binary activation states. The entropy of this distribution measures the variability of a neuron's responses, while the KL divergence evaluates its deviation from a uniform reference, emphasizing neurons with skewed or redundant behavior. Neurons with the highest combined DNR metric, indicating weakly discriminative or overly redundant features, are pruned by updating the pruning mask to exclude them. This method leverages the complementary strengths of entropy and KL divergence to perform a more nuanced evaluation of dying neurons. By pruning entire neurons, the algorithm ensures structural coherence and interpretability of the network, making it particularly well-suited for efficient deployment while maintaining state-of-the-art performance. This advanced pruning logic not only identifies underperforming components but also redistributes capacity more effectively than traditional magnitude-based or random pruning strategies.

## A.2   More Experimental Results

Our experimental focus lies on computer vision tasks. We trained MLPNet, ResNet-18 and VGG-16 networks on MNIST, CIFAR-10 and Tiny-ImageNet. Following the training scheme outlined by Evci et al Evci (2020). For the ResNet and VGG architecture, we expanded the scope of our experiments Rachwan (2022). We used 30G bytes of memory, an NVidia V100 GPU and an Intel(R) Xeon(R) Platinum 8352Y CPU @ 2.20GHz CPU.

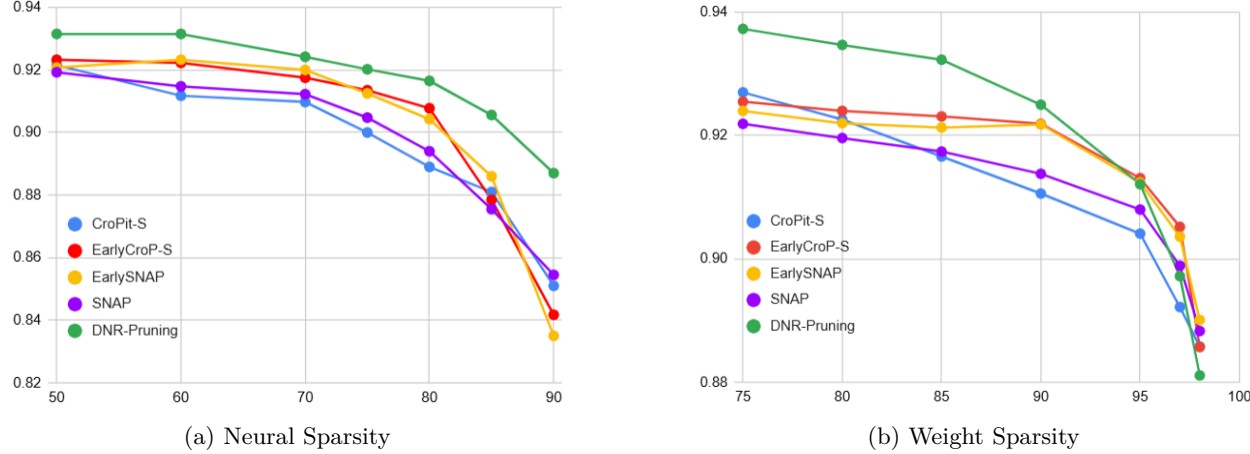

(a) Neural Sparsity         (b) Weight Sparsity

Figure 3: For ResNet-18 networks on CIFAR-10, DNR-Pruning can find sparser solutions maintaining better performance than other approaches. **Left**: Neural sparsity. **Right**: Weight sparsity.

Our approach involves a form of neural "inactivation" or "disabling," where pruning occurs during training, transitioning networks from dense to sparse. The methods we compare against operate within a similar paradigm. We compare our approach with existing methods, including Crop-it/EarlyCrop Rachwan (2022), SNAP Verdenius (2020), and a modified version of the early pruning strategy from Rachwan et al. Rachwan (2022), referred to as EarlySNAP. These methods are trained using the configurations recommended by their authors, without incorporating the regularization regime used in our method. Across all comparisons, our approach achieves comparable or superior performance (Figures 3, 4 and 5).

The results across Figures 3 and 4 demonstrate that DNR-Pruning consistently outperforms other pruning methods by maintaining higher accuracy at equivalent sparsity levels. Specifically, for ResNet-18 on CIFAR-10, DNR-Pruning achieves significantly better performance under both neural and weight sparsity settings (Figure 3), preserving accuracy even at high sparsity levels. This advantage is mirrored in VGG-16 networks (Figure 4), where DNR-Pruning maintains a stable accuracy lead, particularly notable at extreme weight sparsity values. The results confirm that DNR-Pruning's approach to optimizing sparse networks more effectively retains critical model capacity compared to other methods, suggesting that it is better suited to high-sparsity scenarios without substantial loss in accuracy.

---

**Algorithm 1** DNR-Pruning Algorithm

---

**Require:** Training data $\mathcal{D}_{\text{train}}$, validation data $\mathcal{D}_{\text{val}}$, total epochs $T$, learning rate $\eta$, pruning threshold $\tau$, pruning milestones $P$, weighting factors $\alpha$, $\beta$, small constant $\epsilon$
**Ensure:** Pruned model parameters $\theta$
1: Initialize model parameters $\theta$ and mask $M \leftarrow 1$
2: Initialize previous weight distributions $\{W_{i,j}^{(0)}\}$ for all neurons
3: **for** $t = 1$ **to** $T$ **do**
4:     **for** each training sample $(\mathbf{x}, y) \in \mathcal{D}_{\text{train}}$ **do**
5:         $\hat{y} \leftarrow f(\mathbf{x} \mid M \cdot \theta)$
6:         $\theta \leftarrow \theta - \eta \cdot \nabla_\theta \text{loss}(y, \hat{y})$
7:     **end for**
8:     **if** $t \in P$ **then**                                                 ▷ Apply DNR-based pruning
9:         **for** each block $i$ and layer $l$ **do**
10:             **for** each neuron $j$ in layer $l$ **do**
11:                 Compute current weight distribution $W_{i,j}^{(t)}$ from $\theta$
12:                 Compute entropy:

$$H(W_{i,j}^{(t)}) = -\sum_k p(W_{i,j,k}^{(t)}) \log_2(p(W_{i,j,k}^{(t)}) + \epsilon)$$

13:                 Compute entropy change:

$$\Delta H_{i,j} = H(W_{i,j}^{(t-1)}) - H(W_{i,j}^{(t)})$$

14:                 Compute KL divergence:

$$D_{\text{KL}}(W_{i,j}^{(t)} \parallel W_{i,j}^{(t-1)}) = \sum_k p(W_{i,j,k}^{(t)}) \log \frac{p(W_{i,j,k}^{(t)})}{p(W_{i,j,k}^{(t-1)}) + \epsilon}$$

15:             **end for**
16:             Compute $D_{\max} = \max_j D_{\text{KL}}(W_{i,j}^{(t)} \parallel W_{i,j}^{(t-1)})$
17:             **for** each neuron $j$ in layer $l$ **do**
18:                 Compute DNR metric:

$$\text{DNR}_{i,j} = \alpha \cdot \frac{\Delta H_{i,j}}{H(W_{i,j}^{(t-1)}) + \epsilon} + \beta \cdot \frac{D_{\text{KL}}(W_{i,j}^{(t)} \parallel W_{i,j}^{(t-1)})}{D_{\max} + \epsilon}$$

19:                 **if** $\text{DNR}_{i,j} > \tau$ **then**
20:                     $M_j^{(i)} \leftarrow 0$                                             ▷ Prune neuron
21:                 **end if**
22:             **end for**
23:         **end for**
24:         Update $W_{i,j}^{(t-1)} \leftarrow W_{i,j}^{(t)}$ for all neurons
25:     **end if**
26: **end for**
27: **return** $\theta$

---

### A.2.1 Leaky ReLU

Our pruning method is specifically designed for CNNs, where neurons can become entirely inactive, leading to the emergence of dying neurons that can be effectively pruned. While Leaky ReLU Maas (2013) mitigates complete neuron deactivation by allowing small negative activations, it still exhibits a quasi-saturated regime wherein activations remain predominantly negative and contribute minimally to the network's predictions. We

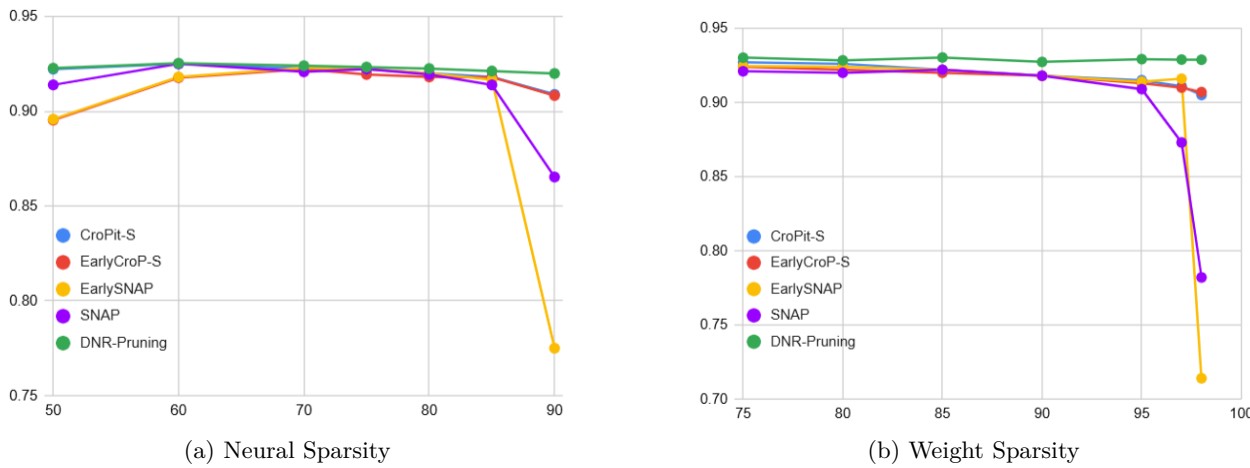

Figure 4: For VGG-16 networks on CIFAR-10. DNR-Pruning better maintains performance at higher sparsities than other approaches. **Left**: Neural sparsity. **Right**: Weight sparsity.

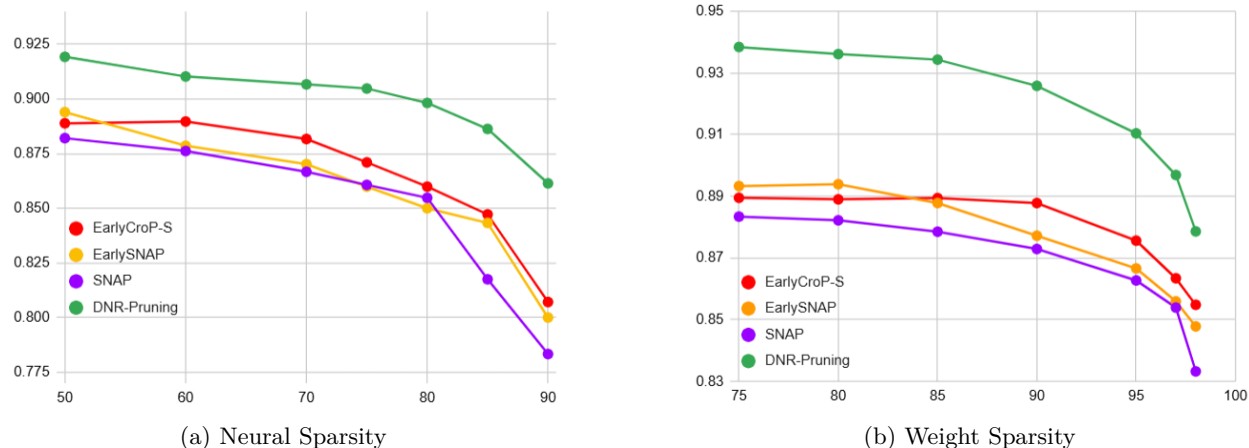

Figure 5: ResNet-18 networks with *Leaky ReLU* trained on CIFAR-10. DNR-Pruning again outperforms the other pruning methods. **Left**: Neural sparsity. **Right**: Weight sparsity.

hypothesize that neurons operating primarily within this low-activation regime behave analogously to dying neurons in ReLU networks, and therefore, they can be pruned without adversely affecting performance. To validate this hypothesis, we extend our approach to ResNet-18 trained with Leaky ReLU activations. In our experiments, we identify and remove neurons that consistently produce negative responses across representative minibatches—indicative of their functional dormancy. As presented in Figure 5, DNR-Pruning consistently outperforms alternative methods across various levels of both neural and weight sparsity, demonstrating its effectiveness even in architectures where neurons do not become completely deactivated.

### A.2.2   Experiment on Tiny-ImageNet

We included results from Pham et al.Pham (2023) in Figure 6 to better illustrate the comparative performance of existing pruning methods (SNIP, Iter-SNIP, SynFlow, PHEW, NPB) against our proposed DNR-Pruning method under varying levels of weight sparsity. The results demonstrate that our DNR-Pruning approach achieves higher accuracy across varying levels of weight sparsity compared to other pruning methods.

Besides weight sparsity, we find that the FLOPs reduction of subnetwork after pruning is more important in the context of pruning before training. We have measured FLOPs of subnetworks produced by different

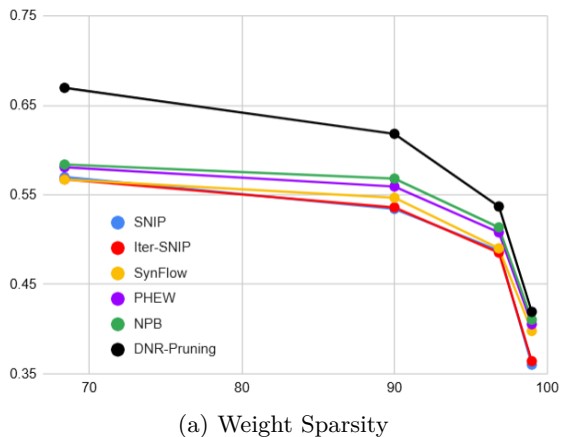

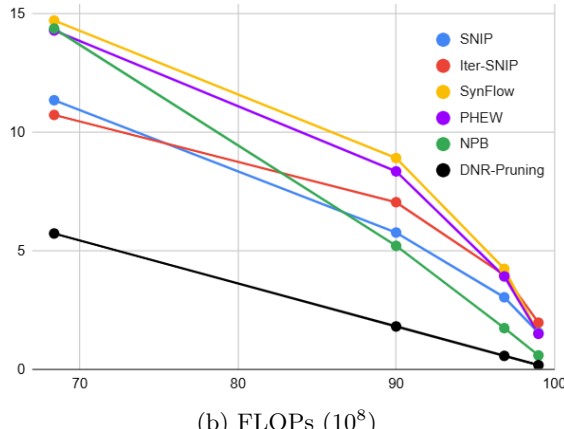

(a) Weight Sparsity

(b) FLOPs ($10^8$)

Figure 6: DNR-Pruning also outperforms other approaches for ResNet-18 networks trained on Tiny-ImageNet. **Left**: Weight sparsity. **Right**: FLOPs($10^8$).

Table 12: For ResNet-20 networks on CIFAR-10, DNR-Pruning again outperforms the other approachs. **Weight sparsity (%)**.

|  | 30 | 40 | 50 | 60 | 70 | 80 | 90 |
|---|---|---|---|---|---|---|---|
| MP | 90.77 | 89.98 | 88.44 | 85.24 | 78.79 | 54.01 | 11.79 |
| WF | 91.37 | 91.15 | 90.23 | 87.96 | 81.05 | 62.63 | 11.49 |
| CBS | 91.35 | 91.21 | 90.58 | 88.88 | 81.84 | 51.28 | 13.68 |
| SSP | 91.37 | 91.19 | 90.60 | 89.22 | 84.12 | 57.90 | 15.60 |
| MSP | 91.25 | 91.20 | 91.04 | 90.78 | 90.38 | 88.72 | 79.32 |
| **DNR-Pruning** | **91.56** | **91.39** | **91.26** | **90.82** | **90.16** | **88.67** | **85.54** |

methods in Figure 6. The result indicates that our DNR-Pruning can produce subnetworks with lower FLOPs than other methods.

### A.2.3  Experiments on MNIST and CIFAR10

The results presented in Tables 6 and 12 (Figure 7) demonstrate the consistent superiority of DNR-Pruning over other pruning methods across varying levels of weight sparsity for MLPNet on MNIST and ResNet-20 on CIFAR-10, respectively. For MLPNet, DNR-Pruning achieves the highest accuracy at all sparsity levels, maintaining robust performance even under extreme sparsity (e.g., 96.72% at 98% sparsity), significantly outperforming other methods Chen (2022a). Similarly, for ResNet-20 on CIFAR-10, DNR-Pruning exhibits the best accuracy at lower sparsity levels and sustains competitive performance as sparsity increases, with a notable margin over alternatives, particularly at higher sparsity thresholds (e.g., 85.54% at 90% sparsity). These results underscore the efficacy of DNR-Pruning in preserving accuracy under constrained model sizes, highlighting its potential for real-world applications where memory and computation are limited.

Table 13: For ResNet-18 networks on CIFAR-10. Comparison of Pruning Interval (PI) with Neural Sparsity (%) of DNR-Pruning.

| PI | 50 | 60 | 70 | 75 | 80 | 85 | 90 |
|---|---|---|---|---|---|---|---|
| 1 | 93.15 | 93.15 | 92.42 | 92.02 | 91.65 | 90.56 | 88.70 |
| 2 | 93.31 | **93.25** | 92.51 | 92.23 | _91.71_ | 90.88 | 88.77 |
| 5 | 93.36 | _93.18_ | 92.56 | 92.17 | 91.69 | 90.73 | 89.14 |
| 10 | _93.56_ | 93.13 | **92.74** | _92.33_ | **91.73** | **90.98** | _89.20_ |
| 15 | **93.56** | 93.12 | _92.64_ | **92.67** | 91.56 | _90.97_ | **89.42** |

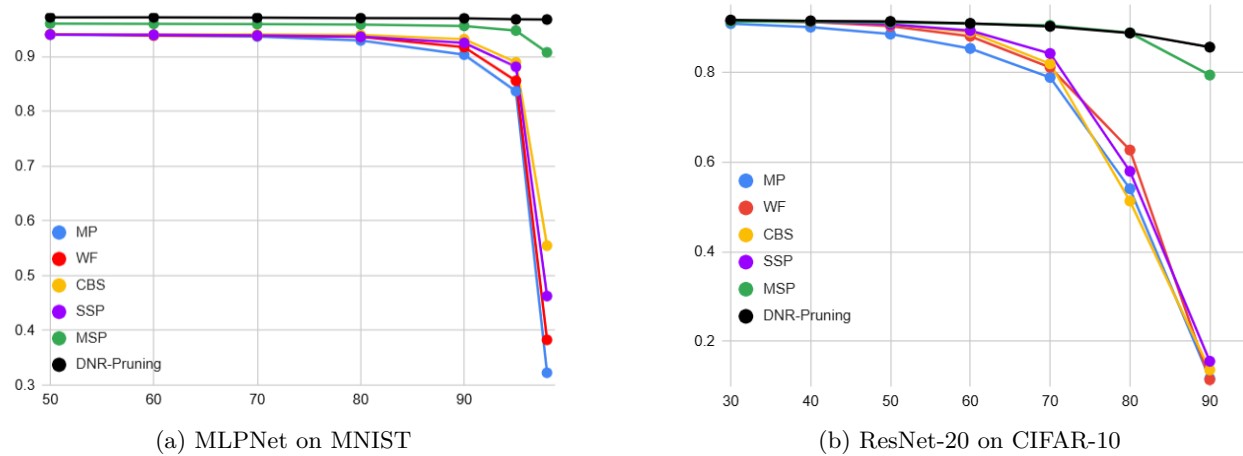

(a) MLPNet on MNIST $\qquad$ (b) ResNet-20 on CIFAR-10

Figure 7: DNR-Pruning also outperforms other approaches for MLPNet and ResNet-20 networks trained on MNIST and CIFAR-10. **Left**: MLPNet on MNIST. **Right**: ResNet-20 on CIFAR-10.

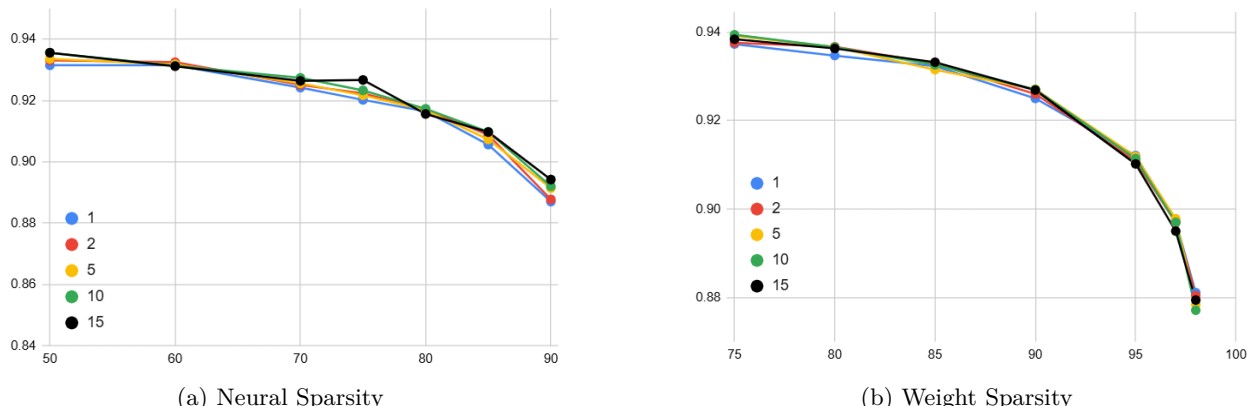

(a) Neural Sparsity $\qquad$ (b) Weight Sparsity

Figure 8: For ResNet-18 networks on CIFAR-10. Comparison of Pruning Interval with different sparsity of DNR-Pruning.

### A.2.4 Pruning Interval

To understand the effect of pruning towards the models during training, Table 13 and 14 (Figure 8) reports the test accuracy of the pruning method with DNR-Pruning. The model accuracy drops after each time parameter trim and gradually recovers to during the non-pruning epochs. As model sparsity increases, the model has a higher impairment on performance and finds it harder to recover. This suggests that the pruning interval can affect the final performance of the network.

Table 14: For ResNet-18 networks on CIFAR-10. Comparison of Pruning Interval (PI) with Weight Sparsity (%) of DNR-Pruning.

| PI | 75 | 80 | 85 | 90 | 95 | 97 | 98 |
|----|------|------|------|------|------|------|------|
| 1 | 93.73 | 93.47 | 93.23 | 92.50 | **91.21** | 89.72 | **88.11** |
| 2 | 93.76 | **93.67** | 93.29 | 92.60 | 91.09 | 89.69 | 88.04 |
| 5 | 93.92 | 93.66 | 93.15 | **92.71** | 91.19 | **89.78** | 87.84 |
| 10 | **93.94** | 93.66 | 93.26 | 92.70 | 91.14 | 89.70 | 87.71 |
| 15 | 93.84 | 93.63 | **93.32** | 92.69 | 91.02 | 89.50 | 87.94 |

Table 13 and 14 (Figure 8) compares the model accuracy given different pruning intervals under different target sparsity. Given sparsity $S$, models are pruned every 1, 2, 5, 10, 15 epochs, where each time the same proportion of parameters is removed from the model. Low pruning interval implies larger amount each time.

For the models pruned by neurons, increasing the pruning interval will result in a slightly higher test accuracy. The model with a pruning interval of 10 and 15 demonstrated a higher test accuracy compared to models with other pruning intervals. But for the weight pruning model, increasing pruning interval can not effectively improve the test accuracy. Specifically, the model with a pruning interval of 10 exhibited higher test accuracy compared to models with other pruning intervals.

### A.3 Codes and Data

Codes, data and more results can be found at `https://github.com/wangbst/DNR-Pruning/`.

