# OpenReview forum: "DNR-Pruning: Sparsity-Aware Pruning via Dying Neuron Reactivation in Convolutional Neural Networks"
_TMLR — Accepted by TMLR_

### Review · Reviewer_3T6f · 2025-04-30

**Summary Of Contributions:**

This paper revisits the phenomenon of dead / dying neurons during deep neural network training, demonstrating that, contrary to the conventional view of them as a hindrance to representation learning, dead neurons can be effectively leveraged to enhance network sparsity through DNR-Pruning. Specifically, the authors propose the DNR pruning score, where a high score indicates that a neuron is experiencing significant entropy reduction alongside increased KL divergence, suggesting that the neuron is progressively becoming redundant. Neurons with high DNR rates will be pruned.


The contributions of this paper are three-fold:

1. The authors empirically study how training parameters (e.g. learning rate, batch size, regularization) affect neuron death.
2. The authors propose DNR-Pruning to achieve dynamic sparsity during optimization.
3. The authors evaluate DNR-Pruning at various sparsity level across different CNN-based neural networks (MLPNet, ResNet-18, VGG-16) and datasets (MNIST, CIFAR-10, Tiny-ImageNet), showing that it outperforms 10 baselines.

**Audience:**

Yes

**Broader Impact Concerns:**

NA.

**Claims And Evidence:**

No

**Requested Changes:**

See above.

**Strengths And Weaknesses:**

Major Concerns

1. To properly validate the effectiveness of DNR-Pruning, experiments on larger-scale models and more recent pruning methods are necessary.

    As noted in Section 4, DNR-Pruning is evaluated against ten baseline methods, many of which are dated and no longer represent the state of the art (e.g., from 2023), using shallow models (e.g., MLPNet, ResNet-18, VGG-16) on small-scale datasets (e.g., MNIST, CIFAR-10, Tiny-ImageNet). Given the significant progress in neural network pruning by 2025, it is essential for the authors to benchmark DNR-Pruning on more challenging settings—such as ImageNet-scale datasets and deeper models like ResNet-50—while also comparing against state-of-the-art baselines published in 2024 and 2025.


2. The modeling of the influence of both weight and input variability in Equations (3) and (4) (Section 3.2) does not account for the nonlinearity of the activation function.

    Equations (3) and (4) did not consider the nonlinearity of $f(\cdot)$. Equation (4) also impose strong assumption that the neuron activations follows a multinomial-like conditional distribution among input channels. However, the rationale of this assumption lacks of justification and discussion. It seems that the entropy of the neuron's output distribution is not used in the DNR pruning score. I am confused of the purpose of introducing this term.

3. The authors are recommended to clarify and justify the distributional assumption imposed on the model weights when introducing $H(W_{i,c}^{(t)})$ in Section 3.2, Page 5.

4. In Lemma 3.5, the authors claim that a high DNR score implies an increasing concentration effect. However, intuitively, a neuron concentrating around a fixed non-zero value does not necessarily indicate that it is dying. Such a neuron may still contribute meaningful, consistent activations and thus should not be considered redundant or prunable solely based on this criterion.

5. The proofs are not included in the manuscript.


-----

Minor Concerns

1. In Section 2, the taxonomy of pruning methods into "sparsity-aware" and "non-sparsity-aware" categories, as proposed in this paper, appears unconventional and potentially misleading.

    The primary distinction among existing pruning methods is more commonly based on whether they perform structured or unstructured pruning, rather than whether they evaluate accuracy variation across different sparsity levels. For example, SynFlow and SNIP are categorized in this paper as "sparsity-aware," yet these methods do not require explicit evaluation of accuracy at varying sparsity levels—they can be directly applied to prune a model to a target sparsity in a single shot.


2. The notations are not consistent. The relation between the lower-cased $w_i$ in equations (2) and (3) and the capitalized $W_{i,c}^{(t)}$ needs to be clarified.

---

> ### Author Response · Authors · 2025-05-28
> **Response to Reviwer#2-Q1**
>
> **R2-Q1:** To properly validate the effectiveness of DNR-Pruning, experiments on larger-scale models and more recent pruning methods are necessary.
>
> **Answer:** We appreciate the reviewer’s point and agree that comprehensive validation on more recent pruning methods and larger-scale models is crucial. **We've add the results in Page 11, Section 4.4, highlighted in red color.** Based on the results in **Table.9 in Page 11**, our proposed method, DNR-Pruning, achieves the highest Top-1 accuracy among all compared pruning approaches on **ResNet-56 with ImageNet, including a recent SOTA method published at WACV
> 2024.** Notably, it reaches this performance within only 90 training epochs, significantly reducing the computational cost compared to methods like Hrank and AdaptDCP. While some methods such as SCP and NuSPM also achieve competitive accuracy, they either lack pre-training or demand longer training schedules. In contrast, DNR-Pruning leverages a dynamic, DNR-based criterion to prune the network effectively at initialization, maintaining high accuracy with minimal training overhead. These results demonstrate the efficiency and robustness of DNR-Pruning in large-scale image classification tasks, highlighting its advantage in both performance and training efficiency.

---

> ### Author Response · Authors · 2025-05-28
> **Response to Reviewer#2-Q2**
>
> **R2-Q2:** The modeling of the influence of both weight and input variability in Equations (3) and (4) (Section 3.2) does not account for the nonlinearity of the activation function.
>
> **Answer:** Thank you to the reviewer for this thoughtful observation. **We've updated this in Page 5, Section 3.2, highlighted in red color.** In deep neural networks, each neuron's activity is characterized not only by its response to input data but also by the evolution of its weight distribution during training. Assuming that the input to a neuron follows a probability distribution
> \begin{equation}
> \(P(x)\) (with \(x = [x_1, x_2, \dots, x_n]\)
> \end{equation}
> representing input features), and that the neuron's activation is given by
> \begin{equation}
> y = f\left( \sum_{j=1}^n w_{i,c,j}^{(t)} x_j + b_{i,c}^{(t)} \right),
> \end{equation}
> this transformation induces an output distribution \(Q(y)\), shaped by both the distribution of pre-activations and the nonlinearity
> \begin{equation}
> \(f(\cdot)\).
> \end{equation}
> The nonlinear function can amplify or suppress variations in the input, making the output entropy a potentially informative proxy of neuron activity. High output entropy suggests dynamic and flexible responses, whereas low entropy implies saturated or inactive behavior, often described as “dying” neurons \cite{Alpher8,Alpher9}.}
>
> To better understand the interplay between input variability and weight magnitude, we define the contribution of each input dimension \(x_j\) to the activation of neuron \(c\) in layer \(i\) at epoch \(t\) as:
> \begin{equation}
> c_j = \left| w_{i,c,j}^{(t)} \right| \cdot \sigma(x_j),
> \end{equation}
> where \(\sigma(x_j)\) denotes the standard deviation of input feature \(x_j\). This formulation emphasizes dimensions with both high input variability and strong weight coupling. A normalized distribution over input contributions is then given by:
> \begin{equation}
> p(x_j) = \frac{c_j}{\sum_{k=1}^n c_k} = \frac{ \left| w_{i,c,j}^{(t)} \right| \cdot \sigma(x_j) }{ \sum_{k=1}^n \left| w_{i,c,k}^{(t)} \right| \cdot \sigma(x_k) }.
> \end{equation}
>
> Although this contribution distribution simplifies the complex relationship between inputs and outputs, it provides a heuristic means to examine the diversity of neuron responses. Based on this, we define the entropy of the neuron's output contribution as:
> \begin{equation}
> H_{\text{out}}^{(t)}(y) = - \sum_{j=1}^n p(x_j) \log_2 \left( p(x_j) \right).
> \end{equation}
> Importantly, this entropy does not directly serve as a pruning score. Instead, it offers conceptual motivation: neurons with low output entropy likely contribute little due to weak input coupling or activation saturation, prompting us to investigate their internal parameter dynamics.
>
> To that end, we track the evolution of each neuron's weight distribution over time. Let
> \begin{equation}
> \(W_{i,c}^{(t)} = \{ w_{i,c,1}^{(t)}, w_{i,c,2}^{(t)}, \dots, w_{i,c,n}^{(t)} \}\)
> \end{equation}
> denote the set of incoming weights to neuron \(c\) in layer \(i\) at epoch \(t\). We define the weight entropy as:
> \begin{equation}
> H(W_{i,c}^{(t)}) = -\sum_{j=1}^n p(w_{i,c,j}^{(t)}) \log_2 \left( p(w_{i,c,j}^{(t)}) + \epsilon \right),
> \end{equation}
> where \(p(w_{i,c,j}^{(t)})\) is a normalized distribution over weight magnitudes and \(\epsilon\) is a small constant added for numerical stability. The change in weight entropy across epochs is defined as:
> \begin{equation}
> \Delta H_{i,c}^{(t)} = H(W_{i,c}^{(t-1)}) - H(W_{i,c}^{(t)}),
> \end{equation}
> which measures how concentrated the weight distribution becomes. Additionally, we evaluate the Kullback-Leibler (KL) divergence between weight distributions at successive epochs:
> \begin{equation}
> D_{\mathrm{KL}}\left( W_{i,c}^{(t)} \parallel W_{i,c}^{(t-1)} \right) = \sum_{j=1}^n p(w_{i,c,j}^{(t)}) \log \left( \frac{p(w_{i,c,j}^{(t)})}{p(w_{i,c,j}^{(t-1)}) + \epsilon} \right),
> \end{equation}
> which captures abrupt shifts in the neuron's weight landscape. A large entropy drop (\(\Delta H_{i,c}^{(t)} > 0\)) combined with a rise in KL divergence signals a reduction in weight diversity—an indicator of potential neuron inactivity or saturation.}
>
> In summary, while the output entropy \(H_{\text{out}}^{(t)}(y)\) highlights the functional stagnation of a neuron due to activation saturation or ineffective connectivity, our pruning strategy is grounded in the temporal dynamics of the weight distribution. This approach allows us to distinguish between structurally stagnant neurons and those with evolving, but currently underutilized, roles.

---

> ### Author Response · Authors · 2025-05-28
> **Response to Reviewer#2-Q3**
>
> **R2-Q3:** The authors are recommended to clarify and justify the distributional assumption imposed on the model weights when introducing \(H(W_{i,c}^{(t)})\) in Section 3.2, Page 5.
>
> **Answer:** We thank the reviewer for raising this important point. **We've added this in Page 19, Appendix Section 1.4, highlighted in red color.**
>
> The computation of weight entropy \(H(W_{i,c}^{(t)})\) relies on interpreting the normalized magnitudes of weights as a discrete probability distribution. Specifically, given a neuron \(c\) in layer \(i\) at epoch \(t\), the weight vector,
> \begin{equation}
> \(W_{i,c}^{(t)} = \{w_{1}, w_{2}, \ldots, w_{n}\}\)
> \end{equation}
> is transformed into a distribution by computing:
> \begin{equation}
> p(W_{i,c,j}^{(t)}) = \frac{|w_j|}{\sum_{k=1}^n |w_k|},
> \end{equation}
> where,
> \begin{equation}
> \(p(W_{i,c,j}^{(t)})\)
> \end{equation}
> can be interpreted as the relative contribution of the \(j\)-th input weight to the overall signal processing of the neuron. This normalization assumes that the absolute magnitudes of weights are meaningful proxies for their relative influence. While this approach does not impose a probabilistic generative model on the weights themselves, it provides a heuristic empirical distribution that facilitates comparison across epochs. The entropy computed from this normalized distribution serves not as a statistical estimation of an underlying latent variable, but rather as a measure of concentration or spread in the weight values.
>
> This assumption is motivated by practical observations in deep learning, where the diversity of incoming weight magnitudes often correlates with functional flexibility. A low entropy suggests domination by a few strong weights or uniform decay (saturation), both indicative of reduced information processing diversity. Thus, the entropy serves as a descriptive tool to identify neurons with concentrated, stagnant, or collapsing connectivity patterns, making it suitable for identifying candidates for pruning under our reactivation-aware framework.

---

> ### Author Response · Authors · 2025-05-28
> **Response to Reviewer#2-Q4**
>
> **R2-Q4:** In Lemma 3.5, the authors claim that a high DNR score implies an increasing concentration effect. However, intuitively, a neuron concentrating around a fixed non-zero value does not necessarily indicate that it is dying. Such a neuron may still contribute meaningful, consistent activations and thus should not be considered redundant or prunable solely based on this criterion.
>
> **Answer:** We thank the reviewer for raising this thoughtful point. **We've modified this in Page 7, Section 3.2, highlighted in red color. And the Lemma Proof added in Page 19, Appendix Section 1.5, highlighted in red color.** While it is true that a neuron with concentrated weights around a fixed non-zero value may still produce consistent activations, such behavior indicates a limited and repetitive encoding pattern. When both entropy decreases and KL divergence increases, the neuron’s weight distribution becomes not just stable but also informationally redundant—i.e., its contribution is predictable or overlapping with other neurons. Therefore, the neuron does not need to be dead to be prunable; redundancy is the core criterion, as supported by experimental evidence.
>
> **Lemma 3.5 (High DNR Indicates Redundant Information Encoding)** (See page 19 in the revised version)
> If, over successive epochs, a neuron \(c\) consistently exhibits a substantial reduction in weight entropy
> \begin{equation}
> (\(\Delta H_{i,c}^{(t)} > 0\))
> \end{equation}
> alongside increased KL divergence—resulting in a high
> \begin{equation}
> \(\text{DNR}_{i,c}^{(t)}\)
> \end{equation}
> —then the neuron's weight distribution becomes increasingly concentrated. This shift implies reduced variability in its representational capacity, suggesting that the neuron's contribution becomes redundant with respect to the learned representation, and can therefore be pruned without loss of expressivity.
>
> **Proof.** Let
>
> \begin{equation}
> \( W_{i,c}^{(t)} = \{ W_{i,c,j}^{(t)} \}_{j=1}^d \)
> \end{equation}
>
> denote the weights of neuron \(c\) in layer \(i\) at epoch \(t\), and let
> \begin{equation}
> \(p(W_{i,c,j}^{(t)})\)
> \end{equation}
> denote an empirical estimate of their distribution. The DNR score is defined as
>
> \begin{equation}
>
> \text{DNR}_{i,c}^{(t)} = \alpha \cdot \frac{\Delta H_{i,c}^{(t)}}{H(W_{i,c}^{(t-1)})+\epsilon} + \beta \cdot \frac{D_{KL}(W_{i,c}^{(t)} \parallel W_{i,c}^{(t-1)})}{D_{\max}+\epsilon},
>
> \end{equation}
>
> where,
> \begin{equation}
> \(\Delta H_{i,c}^{(t)} = H(W_{i,c}^{(t-1)}) - H(W_{i,c}^{(t)}) > 0\)
> \end{equation}
> denotes a drop in weight entropy and
> \begin{equation}
> \(D_{KL}(W_{i,c}^{(t)} \parallel W_{i,c}^{(t-1)}) > 0\)
> \end{equation}
> measures the divergence from the prior distribution. A consistently high DNR score implies that the neuron's weight distribution is becoming increasingly concentrated (i.e., lower entropy), while also undergoing a nontrivial distributional shift (i.e., higher KL divergence), indicating convergent specialization. This leads to reduced variability in the neuron's input-response profile and thus a lower capacity for encoding diverse or novel features. As the neuron's output becomes increasingly predictable and potentially correlated with other similarly specialized units, its representational contribution becomes redundant. Therefore, such neurons—though not necessarily silent—are statistically compressible without performance degradation, and their removal is theoretically justified. The empirical results in our study confirm that pruning high-DNR neurons preserves or even improves accuracy, substantiating DNR as a valid redundancy criterion.

---

> ### Author Response · Authors · 2025-05-28
> **Response to Reviewer#2-Q5**
>
> **R2-Q5:** The proofs are not included in the manuscript.
> **Answer:** Thank you for pointing this out. **We've added this in Page 20, Appendix Section 1.6, highlighted in red color.**
>
> **Proof.** Let
> \begin{equation}
> \(W_{i,c}^{(0)}\)
> \end{equation}
> denote the initial weight vector of neuron \(c\) in layer \(i\), sampled from an arbitrary distribution. The DNR score at epoch \(t\) is defined as
>
> \begin{equation}
> DNR_{i,c}^{(t)} = \alpha \cdot \frac{\Delta H_{i,c}^{(t)}}{H(W_{i,c}^{(t-1)})+\epsilon} + \beta \cdot \frac{D_{KL}(W_{i,c}^{(t)} \parallel W_{i,c}^{(t-1)})}{D_{\max}+\epsilon},
> \end{equation}
>
> where
> \begin{equation}
> \(\Delta H_{i,c}^{(t)} = H(W_{i,c}^{(t-1)}) - H(W_{i,c}^{(t)})\)
> \end{equation}
> and
> \begin{equation}
> \(D_{\max}\)
> \end{equation}
> is the maximum KL divergence across all neurons and training steps. Crucially, both terms are expressed in normalized form: the entropy reduction is divided by the entropy of the previous state, and the KL divergence is scaled by a global maximum. These normalization factors render the metric invariant to the absolute magnitude or scale of the initial weight distribution. More formally, suppose \(W_{i,c}^{(0)}\) is scaled or shifted within a family of distributions; since both entropy and KL divergence are information-theoretic quantities that are invariant under affine-preserving normalizations (up to additive constants canceled by the differencing or ratio), the DNR score remains determined solely by the relative change from \(t{-}1\) to \(t\). Hence, under consistent normalization,
>
> \begin{equation}
> \(DNR_{i,c}^{(t)}\)
> \end{equation}
>
> is unaffected by the specific choice of \(W_{i,c}^{(0)}\), capturing only the dynamics of training-induced structural change.

---

> ### Author Response · Authors · 2025-05-28
> **Response to Reviewer#2-Minor Q1**
>
> **R2-Q1:** In Section 2, the taxonomy of pruning methods into "sparsity-aware" and "non-sparsity-aware" categories, as proposed in this paper, appears unconventional and potentially misleading.
>
> **Answer:** We thank the reviewer for pointing out the conventional structured vs. unstructured taxonomy, and we respectfully argue that our “sparsity-aware” vs. “non-sparsity-aware” distinction captures a complementary dimension of pruning behavior that is orthogonal to structure. **We've updated this in Page 2, Section 2.1, highlighted in red color.**
>
> Specifically, “sparsity-aware” methods—including SynFlow and SNIP—leverage an implicit or explicit evaluation of model sensitivity across multiple sparsity levels to guide mask selection (e.g., by estimating gradients or scores as sparsity varies), whereas “non-sparsity-aware” approaches apply a fixed criterion in a single shot without this dynamic feedback. By categorizing methods along this axis, we highlight the importance of adapting pruning decisions based on how accuracy degrades when progressively removing neurons or weights, which offers deeper insight into pruning resilience and guides our DNR-based design. We clarify this rationale in the revised manuscript to ensure that readers understand that our taxonomy is meant to expose this dynamic evaluation strategy, rather than to replace the established structured vs. unstructured classification.

---

> ### Author Response · Authors · 2025-05-28
> **Response to Reviewer#2-Minor Q2**
>
> **R2-MQ2:** The notations are not consistent.
>
> **Answer:** We acknowledge and appreciate the reviewer’s valuable feedback on this point. **We've updated this in Page 5, Section 3.2, highlighted in red color.** To ensure clarity and consistency in the notation, we make the following definitions:
>
> 1) Let \begin{equation} \( w_{i,c,j}^{(t)} \) \end{equation} denote the weight connecting the \(j\)-th neuron in layer \(i-1\) to the \(c\)-th neuron in layer \(i\) at epoch \(t\).
>
> 2) Let \begin{equation}
> \( W_{i,c}^{(t)} = \{ w_{i,c,j}^{(t)} \}_j \)
> \end{equation}
> represent the full vector of incoming weights to neuron \(c\) in layer \(i\) at epoch \(t\).

---

### Review · Reviewer_vwsw · 2025-05-19

**Summary Of Contributions:**

The authors introduce a neuron pruning metric, DNR, which identifies neurons with low information during training and prunes it, in order to make the network sparser.

**Audience:**

Yes

**Claims And Evidence:**

No

**Requested Changes:**

1. Empirical investigations on the design of the DNR metric as suggested above.

**Strengths And Weaknesses:**

Strengths

1. The introduced neuron pruning metric DNR has been validated on CIFAR10 and Tiny ImageNet.

Weaknesses
1. The authors assumes that neurons having a smaller entropy, which means the output activation is close to ocnstant, is redundant and must be removed. However, this assumption is not justified by the authors. Constant activation neurons can be of importance and have been shown to be useful for retaining or memorizing crucial features, especially in transformers.
2. The motivation of the DNR method is similar to the Wanda method [1] which also measure neuron wise contribution. The use of the cross entropy and the KL divergence term has not been investigated in depth to distinguish the motivation of DNR from other similar works like Wanda.
3. Is the method also proposing reactivation of neurons or is simply a pruning method? In that case, DNR is a misleading name, as neurons are not being reactivated during training, the authors should consider justifying this naming.


[1] Sun, Mingjie, et al. "A Simple and Effective Pruning Approach for Large Language Models." The Twelfth International Conference on Learning Representations.

---

> ### Author Response · Authors · 2025-05-28
> **Response to Reviewer#vwsw**
>
> **R1-Q1:** The authors assumes that neurons having a smaller entropy, which means the output activation is close to constant, is redundant and must be removed. However, this assumption is not justified by the authors. Constant activation neurons can be of importance and have been shown to be useful for retaining or memorizing crucial features, especially in transformers.
>
> **Answer:** Thank you for highlighting this important consideration regarding the potential utility of low-entropy neurons. We agree that constant or near-constant activations can, in some contexts—particularly in architectures like transformers—serve useful roles, such as preserving key features or maintaining memory. \textbf{We've revised this in Page 4, Section 3, highlighted in red color.} To clarify, our method does not assume that all low-entropy neurons are inherently redundant. Rather, we evaluate dying neuron based on a combination of entropy reduction and KL divergence increase. As a result, the pruning process focuses on genuinely inactive or non-contributory neurons. We have added a clearer motivation for this approach. Empirically, our results demonstrate that DNR consistently outperforms SOTA pruning method across diverse CNN architectures and datasets, suggesting that the dying neurons we remove are indeed redundant in practice for these scenarios.

---

> ### Author Response · Authors · 2025-05-28
> **Response**
>
> **R1-Q2:** The motivation of the DNR method is similar to the Wanda method [1] which also measure neuron wise contribution. The use of the cross entropy and the KL divergence term has not been investigated in depth to distinguish the motivation of DNR from other similar works like Wanda.
>
> **Answer:** We thank the reviewer for pointing out the conceptual similarity between our DNR method and the Wanda approach. We agree that both methods share the overarching goal of identifying and removing redundant network components. \textbf{We've updated this in Page 1, Section 1, highlighted in red color.} That said, we would like to clarify important differences in motivation, formulation, and application.
>
> Wanda estimates the importance of weights in large language models (LLMs) by computing the product of weight magnitudes and the norms of the corresponding input activations. This leads to a pruning strategy that operates primarily at the level of individual weights, with an emphasis on per-output pruning tailored to transformer architectures. In contrast, our DNR method focuses on convolutional neural networks (CNNs) and evaluates neuron redundancy through a principled combination of two information-theoretic measures: reductions in entropy and increases in KL divergence. This joint criterion is designed to identify neurons that are both uninformative and misaligned with the data distribution. We believe this dual perspective provides a more nuanced and robust assessment of neuron contribution, particularly for CNNs where spatial and channel-wise dependencies play a critical role. Additionally, while Wanda is specifically engineered for LLMs, DNR is designed with the structural characteristics of CNNs in mind. The architectural and functional differences between these model families necessitate distinct pruning strategies, and our approach reflects these considerations. To further support the distinctiveness and effectiveness of DNR, we provide comprehensive experimental results across a range of CNN architectures and datasets. These demonstrate that DNR consistently achieves competitive or superior performance relative to SOTA pruning methods, highlighting the practical value of our proposed criteria.
>
> In summary, while both DNR and Wanda share a high-level objective, the motivation, theoretical underpinnings, and application domains of our approach are substantially different. We appreciate the opportunity to clarify these distinctions and hope this better contextualizes the contributions of our work.

---

> ### Author Response · Authors · 2025-05-28
> **Response**
>
> **R1-Q3:** Is the method also proposing reactivation of neurons or is simply a pruning method? In that case, DNR is a misleading name, as neurons are not being reactivated during training, the authors should consider justifying this naming.
>
> **Answer:** Thank you for this thoughtful observation. We appreciate the opportunity to clarify the rationale behind the naming of our method. **We've add this in Page 6, Section 3.2.1, highlighted in red color.** While our approach primarily focuses on pruning, the term Dying Neuron Reactivation (DNR) is intended to highlight a key conceptual distinction from conventional pruning techniques. Specifically, DNR does not rely on a static criterion to permanently remove neurons. Instead, it adopts a dynamic perspective—monitoring neuron behavior over time using entropy and KL divergence to identify neurons that appear inactive (or “dying”) at certain stages of training but may later regain importance as the learning dynamics evolve.
>
> In this sense, while we do not explicitly "reactivate" neurons in the traditional sense, our method implicitly allows for reactivation by avoiding premature or permanent pruning decisions. This temporal flexibility is crucial for maintaining the adaptability of the network throughout training.

---

> ### Author Response · Authors · 2025-05-28
> **Response to Reviewer Q4**
>
> **R1-Q4:** Empirical investigations on the design of the DNR metric as suggested above.
>
> **Answer:** We appreciate the insightful observations regarding the design of the DNR metric. In response, we have carefully revised the manuscript to address all relevant concerns. The corresponding updates are clearly highlighted in red in the revised version to facilitate review.

---

### Review · Reviewer_kU5C · 2025-06-17

**Summary Of Contributions:**

The authors propose a new importance metric for pruning neurons in CNNs. Specifically, they define the metric, DNR, by combining weight entropy difference and KL divergence to assess the redundancy of each neuron. Neurons with DNR values below a predefined threshold are pruned. The method is evaluated on various benchmarks, including MNIST, CIFAR, and ImageNet.

**Audience:**

Yes

**Broader Impact Concerns:**

I do not have any concern regarding the ethical implication of the work.

**Claims And Evidence:**

Yes

**Requested Changes:**

I think the current version is generally fine. That said, it would be beneficial to include results on more efficiently optimized architectures, such as MobileNet or EfficientNet, as well as additional ablation studies to better understand the contribution of each component (e.g., entropy and KL divergence).

**Strengths And Weaknesses:**

Pros:
- The explanation and writing are clear and easy to follow.
- The method is evaluated on a comprehensive set of benchmarks.

Cons:
- It would be helpful to see results on more efficient CNN architectures, such as MobileNet or EfficientNet.
- An ablation study showing the individual contributions of the entropy and KL divergence terms would strengthen the analysis.
- I am curious how much the results are sensitive based on the hyperparameters $\alpha$ and $\beta$.

---

> ### Author Response · Authors · 2025-07-09
> **Response to Q1**
>
> **R1-Q1**: *It would be helpful to see results on more efficient CNN architectures, such as MobileNet or EfficientNet.*
>
> **Answer**: We thank the reviewer for raising this valuable point. **We have added the results in Page 10, Section 4.3.2, highlighted in blue in the revised paper.**
>
> Table 1 below presents a comprehensive comparison of Top-1 accuracy across various sparsity levels for MobileNet-V2 on the ImageNet dataset, evaluating several SOTA pruning methods. Notably, our proposed method, **DNR-Pruning**, consistently outperforms all methods across all sparsity regimes. At moderate sparsity, DNR-Pruning achieves the highest accuracy, surpassing strong pruning methods such as UniPTS and RigL. For instance, at **70% sparsity**, DNR-Pruning maintains a Top-1 accuracy of **66.75%**, outperforming UniPTS and RigL. At extreme sparsity levels, where most methods suffer severe degradation, DNR-Pruning still retains significantly better performance, highlighting its superior ability to preserve essential neurons.
>
> These results demonstrate that DNR-Pruning not only achieves SOTA performance under standard pruning ratios but also exhibits strong robustness under extreme compression, validating the effectiveness of our DNR-based selection strategy in identifying neurons critical for generalization.
>
> **Table 1**: *Comparison of pruning methods for MobileNet-V2 on ImageNet. DNR-Pruning achieves the best Top-1 accuracy (%) in comparison to the SOTA methods, including POT, RigL, STR, and UniPTS.*
>
> | Method           | 50%   | 60%   | 70%   | 80%   | 90%   |
> |------------------|--------|--------|--------|--------|--------|
> | POT              | 69.25 | 63.39 | 47.07 | 9.13  | 0.20  |
> | RigL             | 66.57 | 64.75 | 60.72 | 52.19 | 30.44 |
> | STR              | 30.96 | 20.57 | 14.62 | 9.40  | 5.32  |
> | UniPTS           | 69.80 | 68.01 | 64.93 | 59.47 | 42.46 |
> | **DNR-Pruning**  | **70.11** | **68.23** | **66.75** | **62.77** | **46.29** |

---

> ### Author Response · Authors · 2025-07-09
> **Response to Q2**
>
> **R1-Q2**: An ablation study showing the individual contributions of the entropy and KL divergence terms would strengthen the analysis.
>
> **Answer**: We appreciate the reviewer’s insightful suggestion. **We have added the results in Page 12, Section 4.5 (highlighted in blue in the revised PDF).** To evaluate the individual and joint contributions of KL divergence (`α`) and entropy (`β`) in guiding neuron death, we conducted an ablation study on the CIFAR-10 dataset using ResNet-18 under various sparsity levels.
>
> As shown in the table below, pruning solely based on KL divergence or entropy leads to performance degradation as sparsity increases, with KL divergence-only pruning slightly outperforming entropy-only pruning across all settings. Incorporating both signals through a convex combination improves robustness. Notably, the hybrid configurations of `0.75α + 0.25β` and `0.25α + 0.75β` yield competitive but marginally different results, suggesting complementary behavior between the two criteria. However, our proposed DNR-Pruning, which adaptively balances the two signals (i.e., `α = β = 0.5`), consistently achieves the highest test accuracy across all sparsity levels. For instance, at 95% sparsity, DNR-Pruning achieves 91.21%, outperforming KL divergence only and entropy only. These results demonstrate that an equal-weighted fusion of KL divergence and entropy provides a more robust and expressive signal for pruning, validating the effectiveness of our dynamic DNR-Pruning strategy.
>
> **Table: Test accuracy (%) on CIFAR-10 using ResNet-18 under different sparsity levels.**
> _α: KL Divergence, β: Entropy_
>
> | α     | β     | 75%   | 80%   | 85%   | 90%   | 95%   | 97%   | 98%   |
> |-------|-------|-------|-------|-------|-------|-------|-------|-------|
> | 0     | 1     | 93.76 | 93.44 | 93.01 | 92.40 | 90.78 | 89.24 | 87.70 |
> | 0.75  | 0.25  | 93.73 | 93.45 | 93.04 | 92.45 | 90.89 | 89.42 | 87.77 |
> | **0.5** | **0.5** | **93.73** | **93.47** | **93.23** | **92.50** | **91.21** | **89.72** | **88.11** |
> | 0.25  | 0.75  | 93.73 | 93.43 | 93.03 | 92.32 | 90.88 | 89.21 | 87.76 |
> | 1     | 0     | 93.71 | 93.30 | 92.98 | 92.31 | 90.73 | 88.99 | 87.59 |

---

> ### Author Response · Authors · 2025-07-09
> **Response to Q3**
>
> **R1-Q3**: I am curious how much the results are sensitive based on the hyperparameters $\alpha$ and $\beta$.
>
> **Answer**: We appreciate the reviewer’s attention to this important aspect. **We have added the results in Page 12, Section 4.5 (highlighted in blue).** In our previous ablation study (Table 4), we evaluated the sensitivity of our method to the hyperparameters $\alpha$ (for KL divergence) and $\beta$ (for entropy) by varying their relative contributions. The results demonstrate that while both components contribute positively to pruning performance, neither dominates across all sparsity levels. Interestingly, the hyperparameter setting with equal weighting, i.e., $\alpha = \beta = 0.5$, consistently yields the best accuracy under different sparsity settings. This indicates that KL divergence and entropy capture complementary aspects of neuron death, and their balanced integration is crucial to the effectiveness of our DNR-Pruning method.

---

### Decision · Action_Editor_eJ7f · 2025-08-18

**Recommendation:** Accept with minor revision

**Additional Comments:**

Several reviewers noted that the paper lacks theoretical depth. The theoretical explanations in the appendix are more intuitive than rigorous, and Section 3.2 in particular relies on strong assumptions. This is acceptable given that the contribution of the paper is primarily empirical, but the limitations of the theoretical framing should be made more explicit.

For the minor revision, we ask the authors to make the following updates:
- Add a disclaimer to Sections 3.2, A.1.4, A.1.5, and A.1.6 clarifying that these derivations are not fully rigorous but are intended to provide intuition from the authors’ perspective.
- In Table 7, add the accuracy of the unpruned model and include the sparsity of the resulting models.

**Audience:**

Yes

**Audience Explanation:**

Yes. Network pruning and sparsity-aware training are important topics for the TMLR community, given their relevance to efficient deep learning, sustainability, and deployment on resource-constrained hardware.

**Claims And Evidence:**

Yes

**Claims Explanation:**

This paper proposes DNR-Pruning, a sparsity-aware pruning method that leverages a novel metric for identifying and pruning neurons based on their activity across training iterations. The claims are supported by extensive empirical evidence across several architectures (ResNet, VGG, MLP) and datasets (MNIST, CIFAR-10, Tiny-ImageNet, ImageNet). The experiments are clearly described, and additional results were added during the discussion phase, strengthening the evidence.

Most reviewers were convinced by the empirical validation, although one reviewer noted that comparisons to more recent post-training pruning baselines would further strengthen the case. Despite this, the breadth of experiments and clarity of reporting meet TMLR’s criteria for soundness.

---

> ### Author Response · Authors · 2025-09-01
>
> We sincerely thank the reviewers and the editor for their valuable feedback and constructive suggestions, which have greatly helped us improve the quality of our work. We have carefully addressed all the comments and incorporated the suggested minor revisions. The final version of the manuscript has now been uploaded.